# VISUAL SEMANTIC LEARNING VIA EARLY STOPPING IN INVERSE SCALE SPACE

## ABSTRACT

Different levels of visual information are generally coupled in image data, thus making it hard to reverse the trend of deep learning models that learn texture bias from images. Consequently, these models are vulnerable when dealing with tasks in which semantic knowledge matters. To solve this problem, we propose an instance smoothing algorithm, in which the Total Variation (TV) regularization is enforced in a differential inclusion to generate a regularized image path from large-scale (*i.e.*, semantic information) to fine-scale (*i.e.*, detailed information). Equipped with a proper early stopping mechanism, the structural information can be disentangled from detailed ones. We then propose an efficient sparse projection method to obtain the regularized images, by exploiting the graph structure of the Total Variation matrix. We then propose to incorporate this algorithm into neural network training, which guides the model to learn structural features in the process of training. The utility of our framework is demonstrated by improved robustness against noisy images, adversarial attacks, and low-resolution images; and better explainability via visualization and frequency analysis.

## 1 INTRODUCTION

Deep learning models have achieved great success in abundant computer vision tasks like image recognition, detection, and segmentation, through the usage of large-scale image datasets Krizhevsky et al. (2012); Simonyan & Zisserman (2014); Long et al. (2015). As shown by previous works Geirhos et al. (2018b), the neural networks are prone to learn more texture bias from the image data rather than the structural information like shape. On the other hand, studies Brendel & Bethge (2019) have also realized that low-frequency features like shapes and edges can help make models more robust, which means it is also important to learn this kind of feature during model training. However, since texture and shape are generally entangled in real-world data, it is hard to change the tendency of texture bias based on these raw image data.

To see if there is a remedy for this problem, we resort to the concept of total variation (TV) regularization, which has been widely applied in image denoising and stylization Rudin et al. (1992); Chan & Vese (2001); Osher et al. (2005); Chambolle & Pock (2011). Such a TV regularization gives rise to the spatial smoothing prior, *i.e.*, adjacent pixels tend to have the same value. With such a regularization, the noisy information can be smoothed away and only structural information is maintained. Particularly, Burger et al. (2005) propose the *Inverse Scale Space* (ISS) method for image denoising, which progressively learns finer scales as iterates, until a noise-free image is recovered when stops properly.

Inspired by such an inverse-scale-space (ISS) property Burger et al. (2005), we propose a semantic-aware instance smoothing method based on *Splitted Bregman ISS* Huang et al. (2016), which can disentangle semantic/structural information from details. Specifically, our method is guided by a differential inclusion, which can efficiently enforce TV regularization on an augmented parameter introduced by a variable splitting term. With this TV regularization, this differential inclusion enjoys the ISS property in that it can generate a TV-regularized image path, transitioning from a larger scale, associated with structural information, to a finer scale, associated with detailed information. In this regard, we can disentangle the structural information from detailed ones if the image path is stopped at a proper time, as illustrated in Fig. 1. To obtain the TV-sparse estimator, we project onto the sparse subspace by exploiting the connected components of the graph that the TV matrix corresponds to.

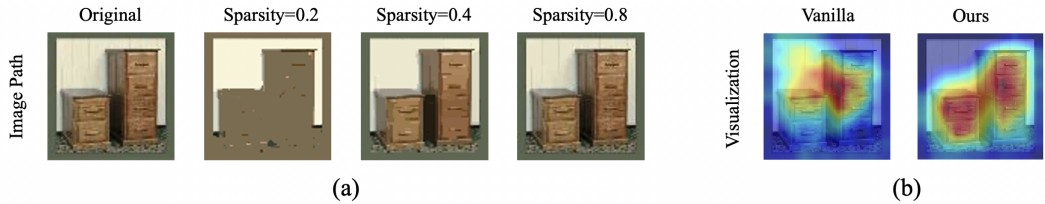

Figure 1: Illustration of Semantic Learning. (a) Image path generated by instance smoothing, less sparsity means less details. (b) Visualization using Grad-Cam. By smoothing details, our method (trained sparsity $= 0.6$) better captures semantic information than vanilla training on original images.

We show that this projection can be efficiently completed in $\mathcal{O}(p)$ ($p$ denotes the dimension of the image vector) time complexity.

To incorporate this algorithm into neural network training, we propose several training procedures, including *fixed training procedure* and *iterative training procedure*. Specifically, the fixed training directly trains the model parameters on smoothed data with fixed sparsity; while the iterative training alternatively runs the instance smoothing algorithm and optimizes the model parameters. Besides, we can also apply the above procedure to tune any trained model. To validate the benefit brought by the proposed pipeline, we conduct extensive experiments among tasks including adversarial attack, low-resolution image classification, noisy images, *etc*. In addition to enhanced robustness in these tasks, we also notice improved interpretability through frequency analysis.

Our main contributions are summarized as follows.

- We propose a novel instance smoothing algorithm that can disentangle the structural information from detailed ones.
- We propose several training procedures that can efficiently incorporate our instance smoothing algorithm into the training procedure.
- Our model achieves promising results on robustness tasks with better explainability.

## 2 RELATED WORK

**Total Variation in Computer Vision** Total Variation (TV), proposed by RUD (1992), has been successfully applied in various vision tasks including denoising Beck & Teboulle (2009); Chambolle (2004), deconvolution Chan & Wong (1998), deblurring Beck & Teboulle (2009), inpainting Afonso et al., superresolution 48 (2008), structure-texture decomposition 6 (2006), and segmentation Donoser et al. (2009). Recently, Yeh et al. (2022a) has shown the benefit for deep learning models brought by the introduction of TV Optimization layer. Different from these methods, we consider the idea of TV-constrained image reconstruction problem in the perspective of semantic-aware learning.

**Linearized Bregman Iteration (LBI)** LBI, a method for solving convex optimization problems, was originally proposed in Osher et al. (2005); Yin et al. (2008). It has been demonstrated that LBI exhibits convergence for convex loss functions, as well as the fundamental properties of discretized differential inclusion dynamics Osher et al. (2016); Huang & Yao (2018). Subsequent research has built upon the LBI framework, introducing various enhancements. In this study, we delve into the practical application of this concept. Specifically, we leverage the inverse scale space property of differential inclusion to address the TV regularization problem, resulting in a versatile solution path for image smoothing.

## 3 SEMANTIC-AWARE LEARNING IN INVERSE SCALE SPACE

In this section, we will introduce our framework for learning semantic features via Inverse Scale Space (ISS). In Sec. 3.1, we will first introduce the instance smoothing method to decompose the semantic and non-semantic information in the inverse scale space, followed by incorporation of this smoothing method to the neural network training in Sec. 3.2.

## 3.1 INSTANCE SMOOTHING IN INVERSE SCALE SPACE

To smooth detailed information for image $x \in \mathbb{R}^p$ ($p := h \times w$ denotes the size of the image vector, with $h, w$ *resp.* denoting the height and width), typically one can enforce the following Total-Variation (TV) regularization Rudin et al. (1992), which has been widely applied in image denoising Rudin et al. (1992); Osher et al. (2005):

$$\mathcal{L}_{\mathrm{TV}}^{\lambda}(\beta) = \frac{1}{2}\|\beta - x\|_2^2 + \lambda\|D\beta\|_1, \tag{1}$$

where $\lambda > 0$ denotes the regularization hyperparameter, and $D \in \mathbb{R}^{m \times p}$ denotes the total variation matrix that corresponds to the graph in which the edge set $E$ (with $|E| = m$) contains those adjacent pairs of pixels, such that $\|D\beta\|_1 := \sum_{(i,j) \in E} |\beta(i) - \beta(j)|$. The TV-regularized image $\beta_\lambda$ is obtained by minimizing this loss. However, solving the solution path of $\beta_\lambda$ w.r.t. $\lambda$ in Eq. 1 is time-consuming Yeh et al. (2022b) since one has to solve Eq. 1 for each $\lambda$. Although several methods have been proposed for acceleration Yeh et al. (2022b); Xin et al. (2014), it is still too expensive to apply in large-scale data.

To efficiently enforce this TV-regularization, we employ the following *Split Bregman Inverse Scale Space (ISS)* Huang et al. (2018; 2016) that was proposed for sparse recovery:

$$0 = -\nabla_\beta \mathcal{L}_v(\beta_t, \gamma_t), \tag{2a}$$
$$\dot{\rho}_t = -\nabla_\gamma \mathcal{L}_v(\beta_t, \gamma_t), \tag{2b}$$
$$\rho_t \in \partial\|\gamma_t\|_1, \tag{2c}$$

where $\mathcal{L}_v(\beta, \gamma) = \frac{1}{2}\|\beta - x\|_2^2 + \frac{1}{2\nu}\|D\beta - \gamma\|_2^2$ denotes the variable splitting term that has been proposed in ADMM Boyd et al. (2011) and Split Bregman Ye & Xie (2011) for implementation convenience. Equipped with such a splitting term, one can enforce sparsity on $\gamma$, the distance of which from $D\beta$ is controlled by the hyperparameter $\nu$. The dynamics in Eq. (2) is a differential inclusion, which generates a regularized image path from large-scale to fine-scale, with $t$ playing a similar role as $1/\lambda$ in Eq. 1. This is because the path $\gamma_t$ transitions from sparsity to density as $t$ increases. Furthermore, in accordance with the ISS property Burger et al. (2005), the non-zero elements of $\gamma_t$ earlier in the process correspond to larger-scale information within the image.

Specifically, we first note from Eq. (2b) that $\rho_t$ follows a gradient descent flow, starting from $\rho_0 = 0$ (hence $\gamma_0 = 0$). As $t$ grows, more elements $\rho_t \in \partial\|\gamma_t\|_1$ tend to hit the boundary of $\pm 1$, making corresponding elements of $\gamma_t$ selected to be non-zeros according to Eq. 2c. With a sparse $\gamma_t$ at each $t$, we can obtain a sparse TV-regularized image $\tilde{\beta}_t$ by projecting $\beta_t$ onto the subspace expanded by the support set of $\gamma_t$, *i.e.*, $S_t := \mathrm{supp}(\gamma_t) := \{i : \gamma_t(i) \neq 0\}$. Due to this projection, we have $D_{S_t^c}\tilde{\beta}_t = 0$, meaning that $\tilde{\beta}_t$ smooth out the information outside $S_t$. Since $\gamma_t$ gets denser (*i.e.*, $S_t$ is larger) as $t$ grows, $\tilde{\beta}_t$ will learn more information. According to the *ISS* property of Split Bregman ISS Burger et al. (2005); Huang et al. (2016), we know that $\tilde{\beta}_t$ will progressively learn finer-scale information as $t$ grows. This means if we stop early in the image path (say $t_0$), then $\tilde{\beta}_{t_0}$ is able to keep only semantic information while more detailed information will be smoothed out.

**Discussions of Split Bregman ISS and Our Specification.** The Split Bregman ISS, which was proposed in the sparse inference of model parameters Huang et al. (2018), and later applied to many machine learning tasks including medical imaging Sun et al. (2017), transfer learning Zhao et al. (2018), and neural network pruning Fu et al. (2020). However, these methods primarily focused on learning important parameters in the model. **In contrast**, we are the first to explore the ISS property at the image level, with the goal of extracting semantic information from the original image.

**Discretization.** To implement, we follow Yin et al. (2008); Huang et al. (2018) to consider a discrete form of Eq. 2, with step size $\alpha$ and the damping factor $\kappa > 0$:

$$\beta_{k+1} = \beta_k - \kappa\alpha\nabla_\beta\mathcal{L}(\beta_k, \gamma_k), \tag{3a}$$
$$z_{k+1} = z_k - \alpha\nabla_\gamma\mathcal{L}(\beta_k, \gamma_k), \tag{3b}$$
$$\gamma_{k+1} = \kappa\mathrm{prox}_{\|\gamma\|_1}(z_{k+1}), \tag{3c}$$

where $\mathrm{prox}_{\|\gamma\|_1}(z_t) := \arg\min_u \frac{1}{2}\|u - z_t\|^2 + \|u\|_1 = \mathrm{sign}(z_t)\max(|z_t| - 1, 0)$. As pointed out in Huang et al. (2016), Eq. 3 will converge to the original dynamics Eq. 2 by letting $\alpha \to 0$ and

$\kappa \to \infty$. Besides, the step size $\alpha$ should satisfy $\alpha < \frac{2}{\kappa \|H_\nu\|_2}$ with $H_\nu := \nabla^2 \mathcal{L}_\nu(\beta, \gamma)$, in order to make $\mathcal{L}_\nu(\beta_k, \gamma_k)$ decrease as iterates. Compared to TV regularization in Eq. 1 that has to run several optimizations, we can easily obtain a regularized image path at all scales with a single run of Eq.3.

**Sparse Projection via Graph Algorithm.** With $\beta_k$ and $\gamma_k$ at each $k$, we can obtain the TV regularized image $\tilde{\beta}_t$ by projecting $\beta_k$ onto the sparse subspace of $\gamma_k$, *i.e.*, $S_k := \mathrm{supp}(\gamma_k)$:

$$\tilde{\beta}_k = \mathrm{proj}_{S_k}(\beta_k) := \arg \min_{D_{S_k^c} \beta' = 0} \|\beta' - \beta_k\|_2, \tag{4}$$

which has a closed-form solution, *i.e.*, $\tilde{\beta}_k = (I - D_{S_k^c}^\dagger D_{S_k^c})\beta_k$[1]. Here $D_{S_k^c}^\dagger$ denotes the pseudo-inverse matrix of $D_{S_k^c}$. The cost is $\mathcal{O}(|S_k^c|^3)$, which is much larger than the cost of gradient descent that is $\mathcal{O}(p)$ when $|S_k^c|$ is large.

To improve the efficiency, we exploit the graph structure of $D_{S_k^c}$. Specifically, note that $D_{S_k^c}$ corresponds to the graph $G := (V, E_{S_k^c})$, such that

$$D_{S_k^c}(\tilde{\beta})(i, j) := \tilde{\beta}_k(i) - \tilde{\beta}_k(j) = 0, \; \forall (i, j) \in E_{S_k^c}.$$

In other words, we have $\tilde{\beta}_k(i) = \tilde{\beta}_k(j)$ if and only if $i$ and $j$ are connected by a path. Inspired by this, we propose to decompose the graph into connected components, such that $\tilde{\beta}_i$ shares the same value in each component. To minimize $\|\tilde{\beta}_k - \beta_k\|_2$, such a value should equal the average of $\beta_k$ in that component. Since the complexity of finding connected components of a $p$-node graph is $\mathcal{O}(p)$, the projection has the same cost as the gradient descent. Our result is summarized as follows.

**Proposition 3.1.** *Given $\beta_k$ and $S_k := \mathrm{supp}(\gamma_k)$ and suppose $G = (V, E_{S_k^c})$ has $C$ connected components $G_1 = (V_1, E_1), ..., G_C = (V_C, E_C)$, such that $V = V_1 \cup ... \cup V_C$, then $\tilde{\beta}_k = \mathrm{proj}_{S_k}(\beta_k)$ can be given by the following with complexity $\mathcal{O}(p)$:*

$$\tilde{\beta}_k(j) = \overline{\beta}_k(V_c), \; \forall j \in V_c \text{ for some } c \in \{1, .., C\}, \text{where } \overline{\beta}_k(V_c) \text{ denotes the average of } \beta_k(V_c).$$

**Extension to colored image via group sparsity.** For a colored image, we have $x \in \mathbb{R}^{p \times 3}$. This means each pixel is a 3-d vector $x_i = [x_{i1}, x_{i2}, x_{i3}]$ in the RGB channels. Correspondingly, we enforce group sparsity on $\gamma \in \mathbb{R}^{p \times 3}$, where each group corresponds to the vector $\gamma_i \in \mathbb{R}^3$:

$$P(\gamma) = \|\gamma\|_{1,2} := \sum_i \|\gamma_i\|_2 = \sum_i \sqrt{\gamma_{i1}^2 + \gamma_{i2}^2 + \gamma_{i3}^2}. \tag{5}$$

By replacing the penalty $\|\gamma\|_1$ with $P(\gamma)$, we can obtain $\gamma_k$ from $z_k \in \mathbb{R}^{p \times 3}$ as follows:

$$\gamma_i = \mathrm{prox}_{\|\gamma\|_{1,2}}(z)_i := \begin{cases} \left(1 - \frac{1}{\|z_i\|_2}\right) & \|z_i\|_2 \geq 1, \\ 0 & \text{otherwise,} \end{cases} \tag{6}$$

which can replace Eq. (3c) to generate the image path for colored images.

## 3.2 INCORPORATION TO THE TRAINING PROCEDURE

In this section, we introduce several strategies to incorporate Eq. (3) into the training procedure: *fixed training*, *iterative training*, and *Finetuning*. By decomposing the semantic and detailed information apart, these training methods are endowed with better interpretability; moreover, they can exploit semantic information to improve robustness against non-semantic perturbation, such as natural noise, high-frequency perturbation, adversarial noise, and low-resolution images.

Specifically, we denote $f_\theta : \mathcal{X} \to \mathcal{Y}$ as the neural network with parameters $\theta$, which is typically trained via *Empirical Risk Minimization* (ERM) with loss $\ell(f_\theta(x), y)$.

**Fixed Training Procedure.** Simply speaking, it means training $f_\theta$ via ERM on regularized image data when stopped at a fixed sparsity level of $\gamma$ (*i.e.*, the proportion of non-zero elements of $\gamma$ over the dimension of $\gamma$). This procedure can be applied to the task of classification with noisy images

---

[1]For a general matrix $A$, we denote $A_S$ as the sub-matrix of $A$ with rows indexed by $S$

and adversarial defense, where the image at an early stopped iteration in Eq. (3) can eliminate the detailed information in which the adversarial perturbation happens.

**Iterative Training Procedure.** Equipped with an efficient generation of the image path, we can iteratively train the network parameter $\theta$ and run the instance smoothing in Eq. (3). In this regard, the model can first learn semantic features, followed by detailed/fine-scale features. Specifically, the iterative training alternatively runs the LBI and the gradient descent w.r.t. $\theta$ as follows:

$$\beta_{k+1} = \beta_k - \kappa\alpha\nabla_\beta\mathcal{L}_\nu(\beta_k, \gamma_k),$$
$$z_{k+1} = z_k - \alpha\nabla_\gamma\mathcal{L}_\nu(\beta_k, \gamma_k),$$
$$\gamma_{k+1} = \kappa * \text{prox}_{\|\gamma\|_1}(z_{k+1}) \text{ from Eq. (3),} \tag{7a}$$

$$\tilde{\beta}_{k+1} = \text{proj}_{\text{supp}(\gamma_{k+1})}(\beta_{k+1}) \text{ from Prop. 3.1,} \tag{7b}$$

$$\theta_{k+1} = \theta_k - \nabla_\theta\ell(f_\theta(\tilde{\beta}_{k+1}, y)) \text{ gradient descent w.r.t. } \theta, \tag{7c}$$

where Eq. (7c) can be replaced with other optimizers such as SGD or Adam. Such an iterative training procedure can decompose the information, which enjoys better interpretability and can be potentially applied to the task when $y$ is labeled according to both semantic and detailed information features, *e.g.*, the medical imaging diagnosis in which both shape and texture of the lesion are pathologically related to the disease. As a compromise, this method may still be open to adversarial attacks since it also contains fine-scale information.

**Finetune Procedure.** For any pre-trained model $f_{\theta_0}$ obtained through a non-regularized training procedure (e.g., vanilla ERM), we perform a fine-tuning process on the parameter $\theta_0$ using the **Fixed Training procedure**, allowing the model to progressively capture semantic information.

## 4 EXPERIMENTS

In this section, we conduct extensive experiments to demonstrate the ability of our method to learn semantic features. We mainly focus on the robustness against non-semantic features introduced by our method. Specifically, the method is evaluated to show the robustness against noisy images, adversarial attacks, high-frequency perturbations, and low-frequency images.

**Datasets.** CIFAR10 Krizhevsky & Hinton (2009) and miniImageNet Vinyals et al. (2016) are adopted in our experiments. For noisy training, we instead utilize CIFAR10-C Hendrycks & Dietterich (2019), which contains different kinds of noisy and corrupted images from CIFAR10.

**Implementation Details.** We use ResNet18 for CIFAR10 and ResNet34 He et al. (2016) for miniImageNet in our experiments. For hyperparameters, we set $\kappa = 10$, $\nu = 1$, and calculate $\alpha$ by $\alpha = \frac{1}{\kappa\|H\|_2}$, where $H = \nabla^2\mathcal{L}_\nu$ is the Hessian matrix of loss function. Since the miniImagenet was originally used for few-shot learning, its classes in the training set and testing set are different. To adapt it into our settings, we split the train set and randomly chose 100 images of each class as our new test set, and others as our training set.

### 4.1 ROBUSTNESS AGAINST NOISY IMAGES

To explain the efficacy of our proposed method when dealing with noisy images, we compare our model with (1) **Vanilla Model**: vanilla training to optimize ERM, and (2) **TV Layer** that appended the neural network with a layer to enforce TV smoothness, following Yeh et al. (2022b). For training, the vanilla model and TV layer are trained on clean images in CIFAR10. For the fixed training procedure, we train the network on preprocessed images from CIFAR10 with sparsity 0.8. For iterative training, we follow Eq. 7 trained with strategy in Eq. 7 with sparsity level from 0.3 to 0.8. For the finetuning procedure, we use fixed training on processed images with sparsity 0.8 to finetune the vanilla model for 20 epochs. In the test stage, we consider three scenarios for both methods: **None**, **Sparsity 0.6**, which respectively correspond to noisy images with no preprocessing, and preprocessing test images via our instance smoothing algorithm in Eq. 3 with sparsity 0.6.

We report the classification accuracy in Tab. 1. When used as a preprocessing method, our method can help almost all the models improve their accuracy on several kinds of noisy data. Meanwhile, our model achieves a further improvement over others by smoothing the detailed information out via preprocessing in the training stage.

Table 1: Classification results on noisy data from CIFAR10-C with different preprocessing strategies.

| Training | Preprocessing on Test Data | Corruption Type | | | | | | | Mean |
|---|---|---|---|---|---|---|---|---|---|
| | | Gaussian | Shot | Impulse | Glass | Motion | Brightness | Elastic | |
| Vanilla Model | None | 45.90% | 59.08% | 51.43% | 55.20% | 78.73% | **93.91%** | 85.72% | 67.14% |
| | Sparsity 0.6 | 72.57% | 76.34% | 51.30% | 57.66% | 78.43% | 92.58% | 84.08% | 73.28% |
| TV Layer | None | 49.97% | 62.14% | 58.75% | 56.84% | **81.27%** | 93.66% | **85.80%** | 69.78% |
| | Sparsity 0.6 | 76.15% | 78.39% | 59.15% | 60.64% | 80.12% | 92.33% | 84.23% | 75.86% |
| **Ours (Fixed Training)** | None | 36.60% | 49.80% | 51.25% | 43.88% | 72.00% | 92.42% | 83.05% | 61.29% |
| | Sparsity 0.6 | 75.34% | 78.01% | 55.03% | 55.80% | 78.08% | 92.47% | 84.38% | 74.16% |
| **Ours (Iterative Training)** | None | 42.90% | 53.55% | 61.54% | 53.19% | 67.99% | 91.01% | 78.94% | 64.16% |
| | Sparsity 0.6 | **78.46%** | **79.83%** | **72.72%** | **62.00%** | 73.72% | 91.15% | 81.03% | **76.99%** |
| **Ours (Finetune)** | None | 42.62% | 55.64% | 52.88% | 49.28% | 75.12% | 93.40% | 83.64% | 64.65% |
| | Sparsity 0.6 | 75.28% | 78.11% | 57.60% | 58.38% | 77.37% | 92.24% | 83.17% | 74.59% |

It is also interesting to observe that instance smoothing yields varying effects with different corruption types. Notably, significant improvements are evident following preprocessing for types like "Gaussian" and "Shot", whereas some other types, such as "Elastic" and "Glass," do not exhibit this phenomenon. To explain, we visualize images corrupted by different types in Fig. 2. As shown, Gaussian or Shot noise mainly corrupts background or contextual details, which can be smoothed out after preprocessing. . In contrast, 'Glass Blur' and 'Elastic Transform' alter shapes significantly, challenging our method's effectiveness.Additionally, 'Brightness' type corruption shows minimal impact, possibly because the noise is relatively not strong.

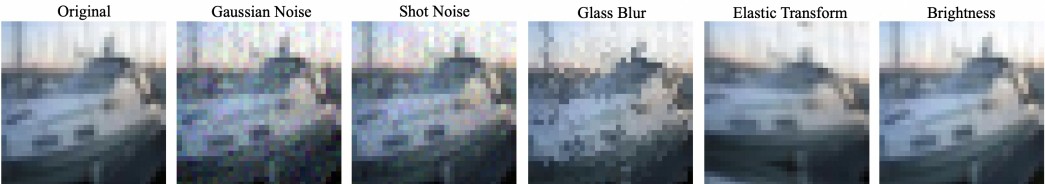

Figure 2: Visualization of different types of noisy images.

## 4.2 ROBUSTNESS AGAINST ADVERSARIAL ATTACK

Table 2: Classification results on adversarial examples (**FGSM**) at different strengths with CIFAR-10.

| Training | Preprocessing on Test Data | $\varepsilon = 8/255$ | $\varepsilon = 16/255$ | $\varepsilon = 24/255$ | $\varepsilon = 32/255$ |
|---|---|---|---|---|---|
| Vanilla Model | None | 26.77 | 18.49 | 15.63 | 14.32 |
| PNI | None | 41.07 | 26.05 | 16.64 | 13.31 |
| TV Layer | None | 43.57 | 31.59 | 21.17 | 16.46 |
| Ours iterative | None | 35.31 | 28.70 | 21.16 | 18.08 |
| Ours fix | None | 37.23 | 26.14 | 18.57 | 13.54 |
| Finetune | None | **48.60** | **36.27** | **23.66** | **17.47** |
| Vanilla Model | Sparsity 0.6 | 38.51 | 31.32 | 27.48 | 25.53 |
| PNI | Sparsity 0.6 | 51.21 | 43.35 | 37.30 | 32.83 |
| TV Layer | Sparsity 0.6 | 53.25 | 45.59 | 40.99 | 36.81 |
| Ours iterative | Sparsity 0.6 | 44.79 | 38.38 | 35.64 | 32.62 |
| Ours fix | Sparsity 0.6 | 51.19 | 43.39 | 38.13 | 33.82 |
| Finetune | Sparsity 0.6 | **57.61** | **51.62** | **46.87** | **41.14** |
| Wang, et al. Natural | - | 17.10 | 14.00 | 12.70 | - |
| Wang, et al. Adv | - | 43.50 | 23.20 | 28.60 | - |

In this section, we show the robustness of our method against adversarial attacks. The attacked data are generated via commonly-used FSGM Goodfellow et al. (2014) and PGD Madry et al. (2018)( In appendix G). We compare our methods with the Vanilla, TV layer methods, PNI Rakin et al. (2018) and results from Wang et al. (2020). For the fixed training procedure, we train the network on

preprocessed images with sparsity 0.6. For iterative training, we follow Eq. 7 trained with strategy in Eq. 7 with sparsity level from 0.3 to 0.6. For the finetuning procedure, we use fixed training on processed images with sparsity 0.6 to finetune the vanilla model for 20 epochs. During the test stage, we consider three scenarios to smooth each data, *None*, *Sparsity 0.6*, and *Sparsity 0.8*, which respectively correspond to test data with no smoothing, and preprocessing with sparsity 0.6 and sparsity 0.8. Since optimizing the TV layer method involves computing the Hessian, which is not computationally tractable for large-scale image data, we have limited its implementation on CIFAR10.

We report the accuracy at strengths $\varepsilon = 2/255$ to $\varepsilon = 8/255$ of CIFAR10 and miniImagenet in Tab. 2 and Tab. 3, where $\varepsilon$ stands for the attack strengths on normalized images. We first note that for all methods, applying the instance smoothing method to test data can bring about robustness improvement, which suggests the utility of instance smoothing. Besides, it is also interesting to see that all variants of our methods can outperform the Vanilla method by a large margin, which can further demonstrate the utility of incorporating the instance smoothing into the training stage. In particular, the finetuning training procedure can outperform the vanilla model by 17.20% at $\varepsilon = 2/255$.

Table 3: Classification results on adversarial examples (**FGSM**) with miniImagent.

| Training | Preprocessing on Test Data | $\varepsilon = 2/255$ | $\varepsilon = 4/255$ | $\varepsilon = 6/255$ | $\varepsilon = 8/255$ |
|---|---|---|---|---|---|
| Vanilla Model | None | 13.23 | 7.86 | 6.17 | 5.30 |
| Ours iterative | None | **16.72** | **9.40** | **6.98** | **6.22** |
| Ours fix | None | 12.84 | 7.39 | 5.84 | 5.08 |
| Finetune | None | 12.20 | 6.78 | 5.14 | 4.47 |
| Vanilla Model | Sparsity 0.6 | 30.54 | 17.58 | 12.75 | 10.39 |
| Ours iterative | Sparsity 0.6 | 30.27 | 16.98 | 12.09 | 9.78 |
| Ours fix | Sparsity 0.6 | 32.62 | 18.67 | 13.09 | **10.48** |
| Finetune | Sparsity 0.6 | **33.92** | **19.81** | **13.55** | 10.31 |

## 4.3 FREQUENCY DOMAIN ANALYSIS

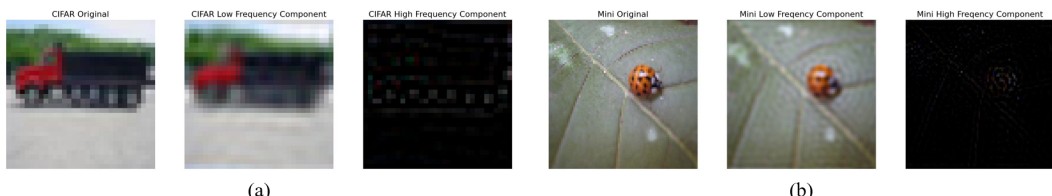

Figure 3: Examples of high and low-frequency components of images. (a) An example from CIFAR10 with a cut-off radius $r = 8$. (b) An example from miniImagenet with a cut-off radius $r = 20$.

To further illustrate the role of our method in enhancing robustness, we try to analyze the method from the perspective of the frequency domain. We follow Wang et al. (2020) to test the accuracy of models on both the high and low-frequency components and follow Geirhos et al. (2018a) to measure the fraction of low-frequency features in our trained model. Specifically, we first decompose the images into low-frequency and high-frequency components as shown in Fig. 3. Then the low-frequency fraction, which is defined as the proportion of correctly predicted instances using only low-frequency components, is calculated among all correctly predicted samples. We consider three kinds of model with different training strategy compared with the vanilla model. For fixed training procedure, we train the network on preprocessed images with sparsity 0.8. For iterative training, we follow Eq. 7 with sparsity level from 0.3 to 0.8. For the finetuning procedure, we use fixed training on processed images with sparsity 0.8 to finetune the vanilla model for 20 epochs.

We plot the low-frequency fraction and accuracy on high/low-frequency components during training with different cut-off radius $r$ in Fig 4 within CIFAR-10 and miniImagenet. As iterates, our method has a higher fraction than the vanilla model. On low-frequency components, our models always achieve the highest accuracy, while the vanilla model usually makes a better prediction on high-frequency components, which is not robust since a human can not get obvious visual information

from high-frequency components. This result suggests the ability of our method to learn semantic information contained in the low-frequency features. Moreover, we note that the iterative training method learns more low-frequency information than the fixed training, which suggests that smoothly increased sparsity in iterative training can facilitate semantic learning.

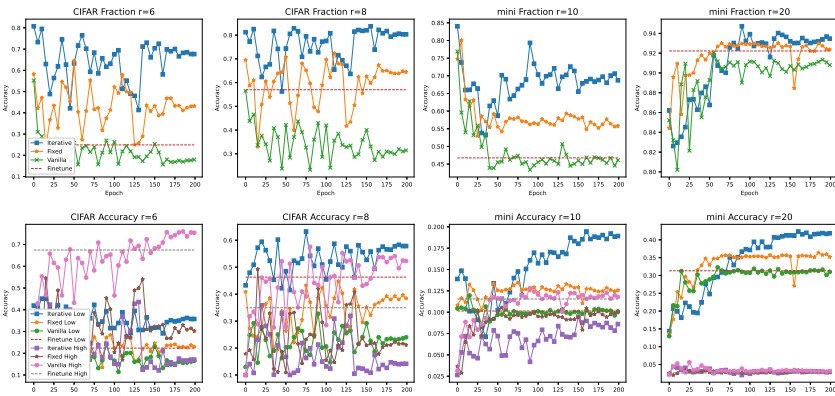

Figure 4: Test accuracy and fraction during training on high/low-frequency components of images from miniImagenet and CIFAR-10. The top row contains a low-frequency fraction on CIFAR10 and miniImagenet. The bottom row contains the accuracy on high/low-frequency components.

Moreover, we present two visualization results. The first one shows the frequencies in the first layer's feature maps during training in Fig. 5. Each grid corresponds to a feature map, where in the high-frequency map is typically more visually dispersed, and the low-frequency map is usually more concentrated. As shown, the vanilla model (bottom) tends to learn high-frequent features while our method can first learn low-frequency features and then high-frequency features in the process of training. This result can explain the low-frequency robustness shown in Fig. 4.

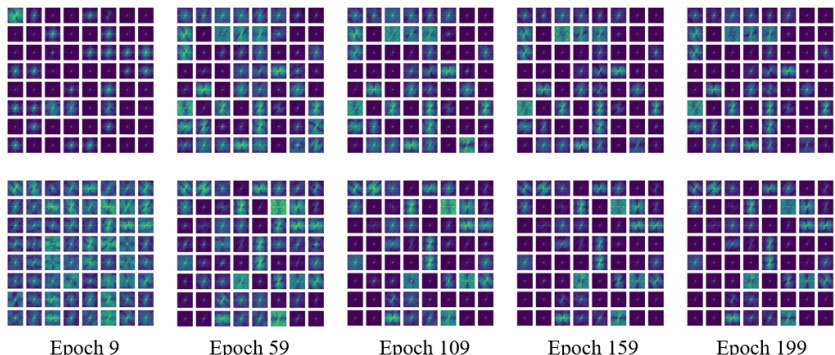

| Epoch 9 | Epoch 59 | Epoch 109 | Epoch 159 | Epoch 199 |

Figure 5: Visualization of the frequency maps in the first convolution layer in the frequency domain during training. The top and bottom rows respectively correspond to our iterative training model and the vanilla method.

The second one is about the expected difference in the frequency domain as proposed in Yin et al. (2019). We calculate $\mathbb{E}(\mathcal{F}(X) - \mathcal{F}(\hat{X}))$, where $\mathcal{F}$ stands for Fourier transformation, $X$ and $\hat{X}$ stand for different images. As shown in Fig. 6, the difference between processed images and the original image is mainly located within the low-frequency component. Besides, as the sparsity level increases from 0.6 to 0.8, the difference in the high-frequency domain between the original images and images generated by our method decreases. These results can explain the low-frequency robustness of our model since during iterative training, the model initially learns low-frequency (large-scale) information and then high-frequency (small-scale) information.

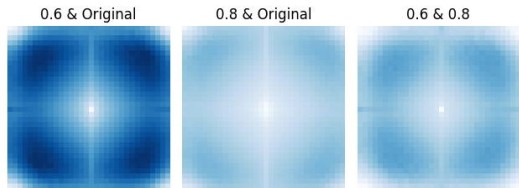

Figure 6: The expected difference in the frequency domain on CIFAR10. "0.6 & Original" stands for the difference between images with 0.6 sparsity and original images. The interpretation of "0.8 & Original" and "0.6 & 0.8" is similar.

## 4.4    ROBUSTNESS AGAINST LOW RESOLUTION

To illustrate the robustness of our method against low-resolution data, we apply our method to the task of classifying low-resolution images. We first downsample the original images to some specific intermediate sizes and then upsample to the original size via nearest interpolation. The smaller intermediate size will result in a lower-resolution image. Similar to the previous section, we consider the fixed model trained on preprocessed images with sparsity 0.6, the iterative model trained with strategy in Eq. 7 from sparsity 0.3 to 0.6 and the finetuned model on preprocessed images with sparsity 0.6.

The results are presented along the training procedure in Fig. 7 for test data with intermediate size sets from 74 to 24 respectively. As shown, all variants of our methods outperform the vanilla model (blue curve), especially with lower-resolution images. This result suggests the effectiveness of instance smoothing in learning semantic information during training, as the low-resolution images can smooth out the details while maintaining the object's shape.

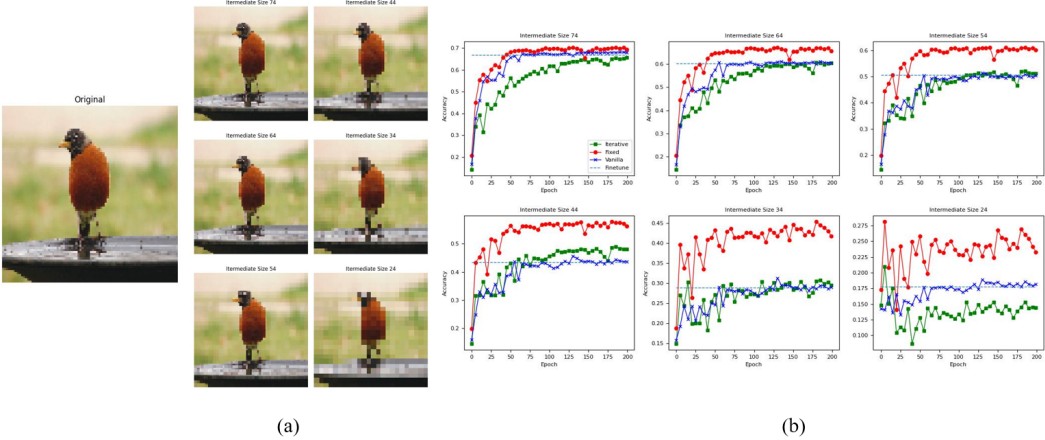

Figure 7: (a) Examples of low-resolution images, with the original image on the left and images of different intermediate sizes from 74 to 24 on the right; (b) Test accuracy during training on low-resolution images with different intermediate sizes.

## 5    CONCLUSIONS AND DISCUSSIONS

We present a novel instance smoothing algorithm that effectively disentangles structural information from images. We propose an efficient graph-based algorithm for projection acceleration. We then propose three procedures to incorporate the algorithm into network training. We demonstrate the utility in several robustness tasks.

**Limitations.** Our methods can bring additional memory costs during training, which makes it difficult to extend to larger-scale datasets such as the Imagenet. Besides, we believe that our method can be potentially applied to feature maps with TV regularization. Such an extension and the optimization of memory usage will be explored in the future.

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

## A   PROOF OF PROPOSITION 3.1

**Notations.** We project $\beta_k$ onto the sparse subspace of $\gamma_k$, *i.e.*, $S_k := \text{supp}(\gamma_k)$: $\tilde{\beta}_k = \text{proj}_{S_k}(\beta_k) := \arg\min_{D_{S_k^c}\beta'=0} \|\beta' - \beta_k\|_2$. We denote $D_{S_k^c}$ as the sub-matrix of $D$ with rows indexed by $S_k^c$, which indexes the nonzero elements of $\gamma_k$. Specifically, $D_{S_k^c}$ is the graph difference matrix of $G := (V, E_{S_k^c})$, such that $(i,j) \in E_{S_k^c}$ if $D_{S_k^c}\tilde{\beta}_k(i,j) := \tilde{\beta}_k(i) - \tilde{\beta}_k(j) = 0$. $E$ is the corresponding edge set of $D$, which is composed of adjacent pairs of pixels.

*Proof of Prop. 3.1.* Suppose $G = (V, E_{S_k^c})$ has $C$ connected components $G_1 = (V_1, E_1), ..., G_C = (V_C, E_C)$, such that $V = V_1 \cup ... \cup V_C$. If two nodes $i$ and $j$ are in the same component, the corresponding elements of $\tilde{\beta}_k$ have the same value, *i.e.*, $\tilde{\beta}_k(i) = \tilde{\beta}_k(j)$. Then for each component $V_c$, $\tilde{\beta}_k(V_c)$ shares the same value. If we denote it as $\eta_c$, then the $\eta_c$ to minimize

$$\sum_{j \in V_c} (\eta_c - \beta_k(j))^2,$$

equals to the average of $\beta_k(V_c)$, *i.e.*, $\text{mean}(\beta_k(V_c))$. Using the strong connected-component algorithm proposed in Lulli et al. (2016), the decomposition of connected components will cost $\mathcal{O}(log(p))$. □

The algorithm is shown in Alg. **??**, and the flowchart of graph algorithm is shown in Fig. 8.

---

**Algorithm 1** Projection by Connected Components in Graph

---

**Input**   : An image $\beta$, current $\gamma_t$, the graph $G(V, E)$ where $V$ denotes the set of pixels and $E$ contains edges defined according to the graph difference matrix $D$ in Eq. (1).

**Output** : $\tilde{\beta}$ via projection in Eq. (4).

1  Find connected components $G_1 := (V_1, E_1), \ldots, G_C := (V_C, E_C)$.

2  For each $i = 1, ..., C$, compute the average of $\beta$ over $V_i$, *i.e.*, $z_i := \sum_{j \in V_i} \beta(j)/|V_i|$ and take $\tilde{\beta}(j) = z_i$ for each $j \in V_i$.

3  **return** $\hat{\beta}$.

---

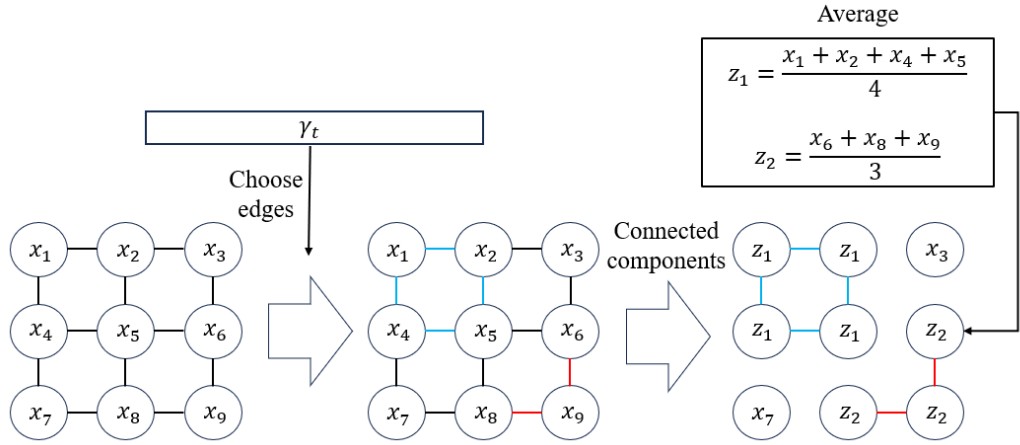

Figure 8: A flowchart of an example to visualize Alg. **??**. There are four components, including $G_3 := \{x_3\}$, $G_4 := \{x_7\}$, and the other components $G_1$ and $G_2$ with vertices in each component connected by blue and red edges. For each component, we take each element's value as the average of $x$ within that component.

*Remark* A.1. After obtaining the connected components, we need to compute the average of each component, which has the complexity of $\mathcal{O}(p)$ and is comparable to the gradient descent. Since the complexity of the soft-thresholding in Eq. (3c) is also $\mathcal{O}(p)$, the overall complexity of our instance smoothing algorithm in Eq. 3 has the same order of the gradient descent.

## B  STANDARD CLASSIFICATION ON CIFAR10 AND IMAGENET100

In this experiment, we apply our method to the standard classification on CIFAR10 and ImageNet100 Deng et al. (2009).

**Experiment Setup.**  We adopt ResNet-18 as the backbone on CIFAR10 and ResNet-50 on ImageNet100. For fixed training, we try different sparsity level for preprocessing images. For iterative training, we use strategy in Eq. 3 from sparsity 0.3 to 0.8.

**Results.**  We report the classification accuracy on CIFAR10 In Tab. 4. As shown, our methods, both fixed and iterative training offer comparable results to the vanilla model after the sparsity reaches 0.6, suggesting that the loss in information is limited. Moreover, fixed training slightly outperforms the vanilla model with sparsity 0.9. We also report the classification accuracy on ImageNet100 In Tab. 5

Table 4: Results of standard classification on CIAFR10.

| Sparsity Level | 0.4 | 0.5 | 0.6 | 0.7 | 0.8 | 0.9 | 1.0 (Vanilla) | Iterative |
|---|---|---|---|---|---|---|---|---|
| Accuracy | $85.24 \pm 1.69\%$ | $92.47 \pm 1.53\%$ | $93.82 \pm 0.68\%$ | $94.53 \pm 0.27\%$ | $94.78 \pm 0.24\%$ | $\mathbf{95.38} \pm 0.14\%$ | $95.29 \pm 0.06\%$ | $94.54 \pm 0.14\%$ |

Table 5: Results of standard classification on ImageNet100.

| Sparsity Level | 0.4 | 0.6 | 0.8 | 1.0 (Vanilla) | Iterative |
|---|---|---|---|---|---|
| Accuracy | $59.12\%$ | $75.51\%$ | $78.66\%$ | $79.36\%$ | $74.39\%$ |

## C  VISUALIZATIONS WITH GRAD-CAM

In this experiment, we apply the Grad-CAM Selvaraju et al. (2019) to visualize learned features during *iterative training*. We consider the model trained with strategy in Eq. 2 from sparsity 0.3 to sparsity 0.8. As shown in Fig. 9, the features learned by our model in the early epochs are more concentrated on the class-dependent regions (*e.g.*, the cat's face in the top-left image and the dog's body in the bottom left image). As iterates, finer-scale information is learned; thus the feature map is enlarged due to the completeness of information. The larger saliency map of our model shows that our model learn more shape information than the vanilla model.

## D  RUNNING TIME OF THE ALGORITHM

In this experiment, we compare the running time of our algorithm on gray-scale images from miniImagenet dataset to sparsity level 0.6. We consider the matrix factorization method and our graph method for the sparse projection in Eq. 7b. We run this test on an NVIDIA Tesla V100 (32GB) and an Intel Gold 6240 CPU @ 2.60GHz.

**Results.**  For other methods such as Singular value decomposition (SVD) decomposition or QR decomposition that can obtain the closed-form solution suffer from high computational cost. Assume $p$ is the dimension of $\beta_k$. For example, the complexity of SVD decomposition in our case is $O(p^3)$, which is much more expensive than the gradient descent. In contrast, the complexity of the graph projection is only $O(p)$. To illustrate, we compare our graph projection methods with other alternatives and also the gradient descent in terms of time complexity. We report the running time for 15,000 iterations on a 84x84 grayscale image, in Tab 6. As shown, our graph projection method is much more efficient than others.

## E  VISUALIZATIONS OF IMAGE PATH

In this experiment, we visualize the regularized image path of more instances on the ImageNet Dataset Deng et al. (2009) and COCO Dataset Lin et al. (2014), a multi-object image dataset .

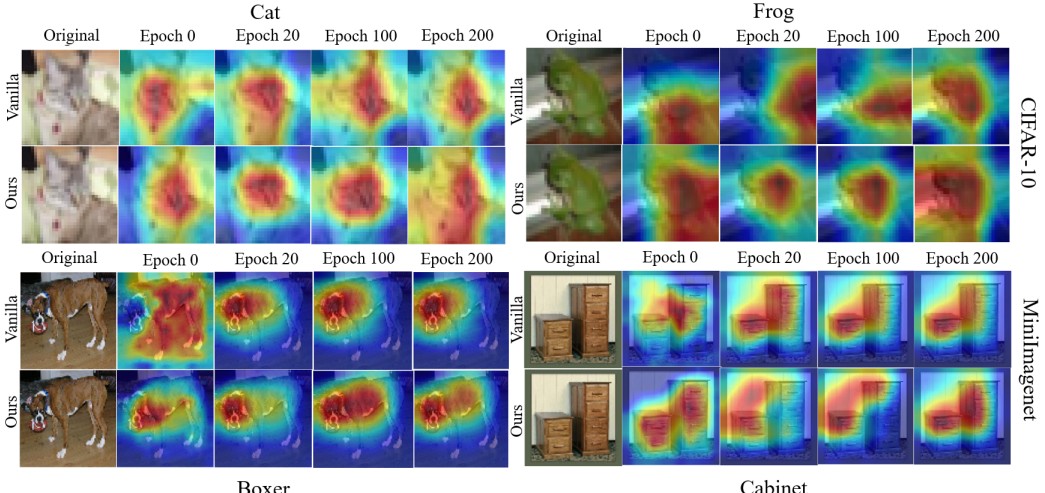

Figure 9: Visualization of learned features in four images: cat (top-left), Boxer (bottom-left), Frog (top-right), and Cabinet (bottom-right) during *iterative training*. The top two are from CIFAR-10 and the bottom two are from miniImagenet. In each image, the top and the bottom rows respectively correspond to the vanilla model and our method in Eq. (7).

Table 6: Computational time ($s$) of different methods for 15,000 iterations on a 84x84 grayscale image.

| Projection Method | SVD | LSQR | Graph Algorithm |
|---|---|---|---|
| Running Time (s) | $373.24 \pm 1.41$ | $171.79 \pm 2.31$ | $4.87 \pm 0.09$ |

**Results.** As shown in Fig. 10, as the sparsity level increases, the image first identifies semantic information and then detailed information. Such semantic information can refer to the shape of the object in the first three rows where the object as a whole has a convex and smoothed boundary; while in the last three rows with irregular and complex contour, such semantic information can refer to the key parts of the object, *e.g.*, the plow of a plow truck in the fifth row, and umbrellas in the last row.

As shown in Fig. 11, when our method meets multi-object images, the shape of the object in the images will pop out when in the beginning of the image path, and more detail texture will gradually add to the background and object smoothly.

## F    TOTAL VARIATION OF CONVOLUTION KERNEL

In this experiment, we calculate the total variation of filters of ResNet18 and visualize the results.

**Experiment Setup.** For training, the vanilla model is trained on clean images of CIFAR10; while our fixed training model is trained on preprocessed images with sparsity 0.8 from CIFAR10; our iterative training model is trained with strategy in Eq. (7) from sparsity 0.3 to 0.8. Denote the kernels of the first convolution layer of ResNet18 as $w_i \in \mathbb{R}^{3 \times 3}$, $i \in \{1, \ldots, 64\}$. We compute the total variation $\|Dw\|_1$ where $D$ corresponds to the total variation (TV) matrix of the image.

**Results.** We plot the histogram of the first layer's kernels' TV in Figure 12. As shown, the kernels both *fixed training* and *iterative training* are more concentrated to smaller TV values than the vanilla model. This can explain the low-frequency robustness shown results in 4.3 that our models have higher low-frequency fraction and tend to extract more low-frequency information.

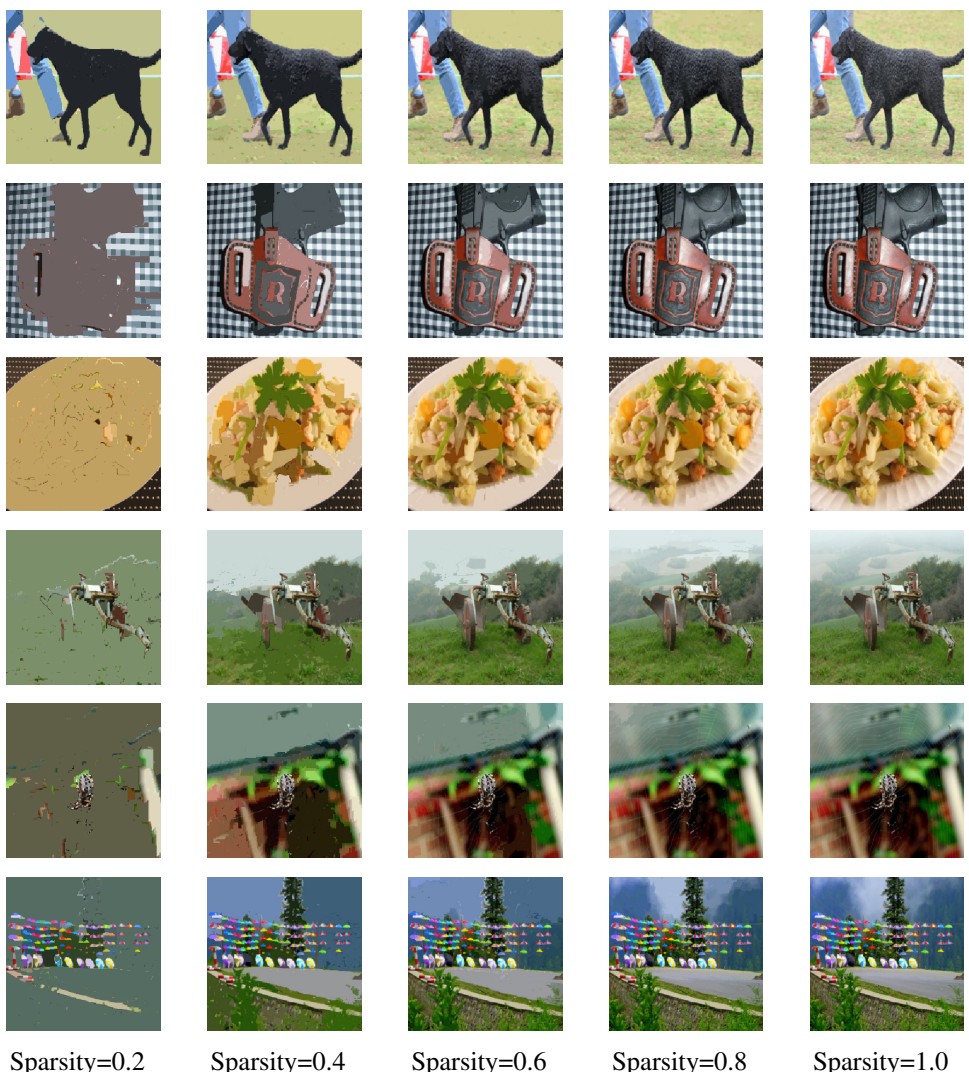

|             |             |             |             |             |
|:-----------:|:-----------:|:-----------:|:-----------:|:-----------:|
| Sparsity=0.2 | Sparsity=0.4 | Sparsity=0.6 | Sparsity=0.8 | Sparsity=1.0 |

Figure 10: The image path generated with our instance smoothing algorithm in Eq. 3. From left to right, the images correspond to sparsity levels of 0.2, 0.4, 0.6, 0.8, and 1.0 (the original image). The 1st to the 6th rows represent a curly-coated retriever; a holster, a dish made of zucchini; a garden spider; a plow; umbrellas.

## G    MORE RESULTS OF ADVERSARIAL ROBUSTNESS

In this section, we present additional results of adversarial examples generated via PGD Madry et al. (2018) in 7 and 8, supplementing Section 4.2.

## H    MORE RESULTS OF FREQUENCY ANALYSIS

In this section, we show more experiment results of the frequency analysis in Sec. 4.3.

**Experiment Setup.** We launch some extra frequency domain analysis experiments on images with more cut-off radius following the settings of high/low frequency components accuracy test in Sec. 4.3. For models, we consider ResNet18 for CIFAR-10 and ResNet34 for miniImagenet. In this part, we only display their accuracy of the last epoch. For training, we consider model trained on preprocessed images on sparsity 0.8, model trained with strategy in Eq. 7 from sparsity 0.3 to 0.8 and model finetuned with preprocessed images with sparsity 0.8.

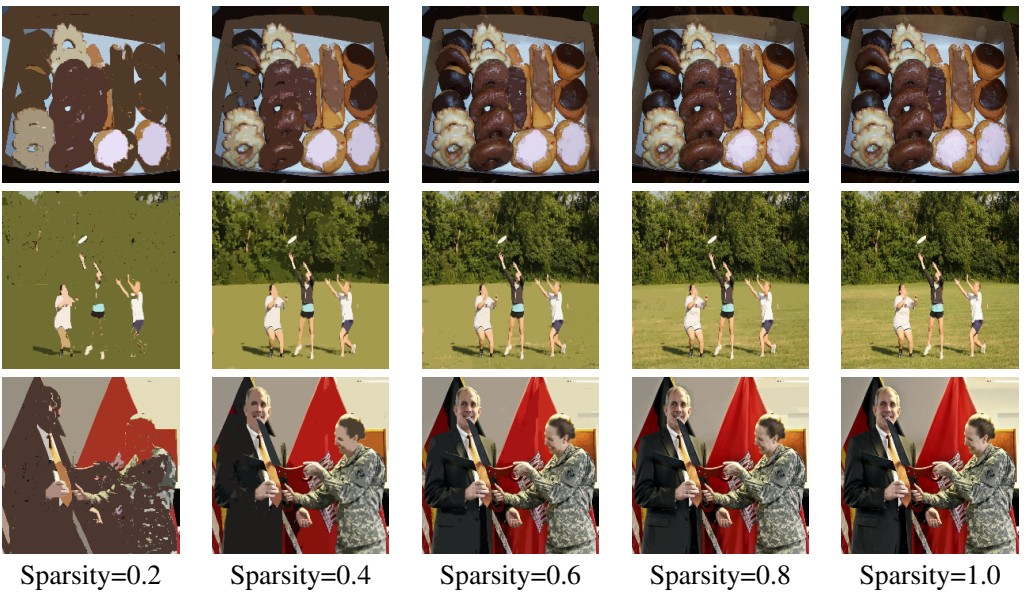

| Sparsity=0.2 | Sparsity=0.4 | Sparsity=0.6 | Sparsity=0.8 | Sparsity=1.0 |

Figure 11: The image path generated with our instance smoothing algorithm in Eq. 3 for multi-object images (COCO dataset).

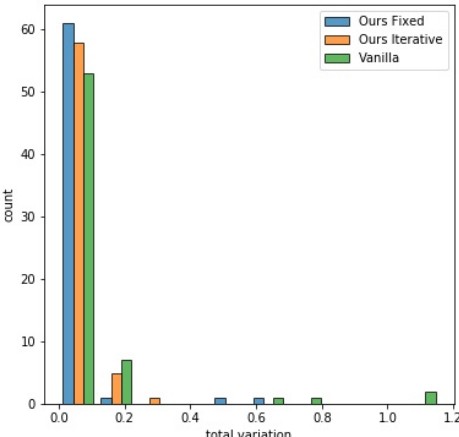

Figure 12: Histogram of the total variation of the first layer's kernels of ResNet18.

Table 7: Classification results on adversarial examples (**PGD**) at different strengths with CIFAR-10.

| Training | Preprocessing on Test Data | $\varepsilon = 8/255$ | $\varepsilon = 16/255$ | $\varepsilon = 24/255$ | $\varepsilon = 32/255$ |
|---|---|---|---|---|---|
| Vanilla Model | None | 0.06 | 0.01 | 0.00 | 0.00 |
| PNI | None | 0.16 | 0.10 | 0.11 | 0.03 |
| TV Layer | None | 0.20 | 0.04 | 0.01 | 0.02 |
| Ours iterative | None | 0.07 | 0.02 | 0.00 | 0.00 |
| Ours fix | None | **1.15** | 0.06 | 0.01 | 0.00 |
| Finetune | None | 0.80 | **0.14** | **0.13** | **0.05** |
| Vanilla Model | Sparsity 0.6 | 29.50 | 29.07 | 32.64 | 34.31 |
| PNI | Sparsity 0.6 | 39.05 | 35.70 | 40.81 | 42.73 |
| TV Layer | Sparsity 0.6 | 39.28 | 35.82 | 39.21 | 41.80 |
| Ours iterative | Sparsity 0.6 | 28.71 | 27.40 | 31.92 | 34.21 |
| Ours fix | Sparsity 0.6 | 38.90 | 34.47 | 38.53 | 41.90 |
| Finetune | Sparsity 0.6 | **41.19** | **38.94** | **43.60** | **43.28** |
| Wang, et al. Natural | - | 8.60 | 7.80 | 7.80 | - |
| Wang, et al. Adv | - | 40.30 | 17.50 | 13.10 | - |

Table 8: Classification results on adversarial examples (**PGD**) at different strengths with miniImagent.

| Training | Preprocessing on Test Data | $\varepsilon = 2/255$ | $\varepsilon = 4/255$ | $\varepsilon = 6/255$ | $\varepsilon = 8/255$ |
|---|---|---|---|---|---|
| Vanilla Model | None | 5.60 | 0.52 | 0.14 | 0.06 |
| Ours iterative | None | **9.38** | 1.08 | **0.28** | **0.16** |
| Ours fix | None | 6.36 | **7.50** | 0.19 | 0.09 |
| Finetune | None | 5.50 | 5.47 | 0.09 | 0.03 |
| Vanilla Model | Sparsity 0.6 | 29.20 | 15.22 | 9.86 | 7.81 |
| Ours iterative | Sparsity 0.6 | 28.17 | 11.80 | 6.75 | 4.44 |
| Ours fix | Sparsity 0.6 | 32.09 | 16.62 | 10.53 | 8.34 |
| Finetune | Sparsity 0.6 | **33.75** | **17.79** | **11.75** | **9.23** |

**Results.** As shown in Tab. 9 and Tab. 10, on both datasets and all the cut-off radius settings, our iterative model can outperform other model on low frequency components and our models always achieve the best low frequency fraction. It show that our model can learn more robust low frequency information, which suggests the effectiveness of the instance smoothing in learning semantic information during training, focusing on more robust low frequency information.

Table 9: Test accuracy and low frequency fraction of last epoch of models on high/low-frequency components of images from CIFAR-10. The $r$ stands for the cut-off radius in frequency domain.

| Model | r=4 | r=6 | r=8 | r=10 | Freq |
|---|---|---|---|---|---|
| Vanilla | **86.11%** | **75.43%** | **52.27%** | **27.77%** | |
| Finetune | 83.23% | 67.48% | 34.95% | 17.96% | High |
| Fixed | 40.99% | 30.24% | 21.16% | 15.91% | |
| Iterative | 22.17% | 17.09% | 14.17% | 13.06% | |
| Vanilla | 11.32% | 16.53% | 23.99% | 46.19% | |
| Finetune | 18.32% | 22.37% | 46.31% | 74.36% | low |
| Fixed | 18.56% | 23.01% | 38.41% | 68.29% | |
| Iterative | **24.79%** | **35.76%** | **57.83%** | **82.02%** | |
| Vanilla | 11.62% | 17.97% | 31.46% | 62.46% | |
| Finetune | 18.04% | 24.89% | 56.99% | 80.55% | Fraction |
| Fixed | 31.16% | 43.21% | 64.48% | 81.11% | |
| Iterative | **52.79%** | **67.66%** | **80.32%** | **86.26%** | |

# I   MORE RESULTS OF LOW RESOLUTION CLASSIFICATION

In this section, we show more detailed results of the low resolution classification task in Sec. 4.4.

**Experiment Setup.** Following the settings of the experiment in Sec. 4.4, we display the classification accuracy of the last epoch of the models on low resolution images from miniImagenet. For models, we consider the model trained on preprocessed images with sparsity 0.6, model trained with strategy

Table 10: Test accuracy and low frequency fraction of last epoch of models on high/low-frequency components of images from miniImagenet. The $r$ stands for the cut-off radius in frequency domain.

| Model | r=10 | r=20 | r=30 | r=40 | Freq |
|---|---|---|---|---|---|
| Vanilla | **11.79%** | **3.14%** | 1.66% | **1.55%** | |
| Finetune | 11.54% | 2.64% | **2.68%** | 0.48% | high |
| Fixed | 9.95% | 2.91% | 1.95% | 1.54% | |
| Iterative | 8.61% | 2.92% | 2.47% | 2.53% | |
| Vanilla | 10.09% | 31.02% | 51.23% | 64.27% | |
| Finetune | 10.14% | 31.31% | 48.70% | 60.14% | low |
| Fixed | 12.56% | 35.20% | 53.29% | 64.83% | |
| Iterative | **18.92%** | **41.84%** | **56.99%** | **65.83%** | |
| Vanilla | 46.11% | 90.81% | 95.87% | 97.65% | |
| Finetune | 46.77% | 92.22% | 94.78% | **99.20%** | Fraction |
| Fixed | 55.79% | 92.37% | **96.46%** | 97.65% | |
| Iterative | **68.73%** | **93.47%** | 95.85% | 96.29% | |

in Eq. 7 from 0.3 to 0.6, and model finetuned on preprocessed images with sparsity 0.6. We consider miniImagenet dataset with different intermediate size.

**Results.** As shown in Tab 11, our fixed training model can outperform the other models on images with different intermediate size. Especially, when the intermediate size is 44, our fixed training model can surpass the vanilla model for 13% in accuracy, demonstrating that our instance smoothing algorithm can help model learn more robust semantic information.

Table 11: The accuracy on low resolution images with different intermediate size. The $s$ stands for the intermediate size.

| Model | s=24 | s=34 | s=44 | s=54 | s=64 | s=74 |
|---|---|---|---|---|---|---|
| Vanilla | 18.14% | 28.98% | 43.64% | 50.37% | 60.51% | 67.93% |
| Finetune | 17.74% | 28.85% | 43.39% | 50.59% | 60.24% | 66.74% |
| Iterative | 14.43% | 29.45% | 48.15% | 51.14% | 60.34% | 65.73% |
| Fixed | **23.29%** | **41.74%** | **56.25%** | **60.18%** | **65.70%** | **69.26%** |

## J    GRAY SCALE LOW RESOLUTION CLASSIFICATION ON MINIIMAGENET

In this section, we further apply our method to the task of classification with low-resolution images.

**Experiment Setup.** Different from the experiment settings in Sec. 4.4, we consider gray-scale miniImagenet data Vinyals et al. (2016) and ResNet34 model with one input channel. We follow the low resolution image generation strategy in Sec. 4.4. To illustrate, Fig. 13 shows an example from gray-scale miniImagenet with different intermediate sizes. For simplicity, we only adopt the iterative training strategy in Eq. (7) from sparsity 0.3 to 0.8.

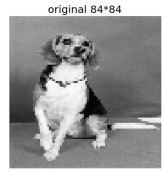 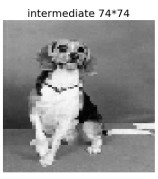 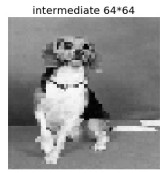 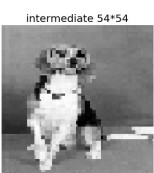

Figure 13: Examples of gray scale low-resolution images intermediate sizes $74 \times 74$, $64 \times 64$, and $54 \times 54$ from left to right. The original size is $84 \times 84$.

**Results.** We present the test accuracy along the training procedure in Fig. 14 for test data with intermediate size set as 74, 64, and 54 respectively. As shown, our method (orange curve) can outperform the vanilla model (blue curve), especially with lower-resolution images. This results are consistent with those in Sec. 4.4 and Sec.I, which also suggests the effectiveness of our instance smoothing algorithm in learning semantic information during training, as the low-resolution image can smooth out the details while maintaining the object's shape. The results show that our model is also robust on gray scale low resolution images, expanding its usage.

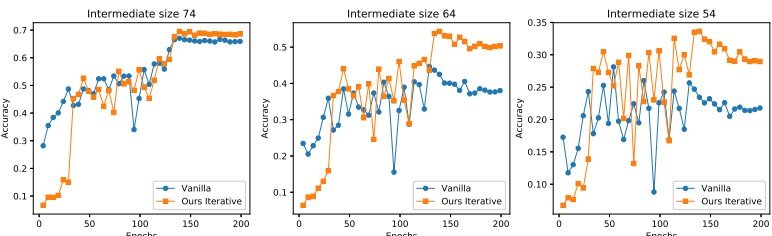

Figure 14: Test accuracy during training on low-resolution images with different intermediate sizes. Left: size 74; center: size 64; right: size 54. Our iterative training in Eq. (7) and the vanilla training are respectively marked by orange and blue.

## K    DENOISED RESULTS ON CIFAR10

In this experiment, we apply our method to image denoising on CIFAR10.

**Experiment Setup.** We implement our instance smoothing algorithm in Eq. 3 to compress the data with sparsity 0.6. We adopt Peak signal-to-noise ratio (PSNR) Hore & Ziou (2010) in Eq. (8), which has been applied to measure the image quality, to evaluate the denoising ability. The Peak signal-to-noise ratio (PSNR), where $X$ denotes the original image and $\tilde{X}$ denotes the noisy images. Here we add Gauss noise with 0 mean and 0.1 standard deviations.

$$\text{PSNR} = 10 \log_{10} \left( \frac{\text{MAX}^2}{\text{MSE}} \right) \tag{8a}$$

$$\text{MAX} = \max X \tag{8b}$$

$$\text{MSE} = \frac{1}{hwc} \sum_{h,w,c} (X - \tilde{X})^2, \tag{8c}$$

where $h, w, c$ denotes the height, width, and number of channels. We visualize the psnr of all the test data on CIFAR10.

**Results.** Fig. 15 shows that the denoised images via our instance smoothing algorithm have larger PSNR than those noisy images. The mean PSNR on the original noisy images is $19.78dB$ but increased to **22.55**$dB$ after denoising. There results demonstrate the effectiveness of our instance smoothing algorithm in image denoising, and explain the results in Sec 4.1 and Sec 4.2.

## L    CLASSIFICATION ON DENOISED DATA FROM IMAGENET

In this experiment, we apply our method to noisy classification of images from ImageNet Deng et al. (2009) with Gaussian noise.

**Experiment Setup.** We apply the pre-trained ResNet 50 on clean data to the test data with Gaussian noise, in which the standard deviations (std) range from 0.01, 0.05, to 0.1. We compared the vanilla model (without preprocessing) and our method that preprocessed the test data via our instance smoothing algorithm in Eq.3 with sparsity 0.6. Due to the limit of time and device, we randomly sample 100 data for testing. To remove the randomness, we repeat it 10 times.

**Results.** Tab. 12 shows that the our instance smoothing algorithm can improve the classification accuracy by $15\%$ across all standard deviations, which suggests the effectiveness of our our instance smoothing algorithm as a preprocessing method to help improve robustness against noise.

**PSNR Visualization.** To further explain the effectiveness of denoising, we visualize the PSNR like Sec. K on each pool of 100 test data with $std = 0.1$ in Tab. 12. As shown, the denoised images via our instance smoothing algorithm have larger PSNR than those of the vanilla model.

**Visualizations of Denoised Images.** To illustrate the effect of denoising, we visualize the denoised image in Fig. 17. As shown, the denoise images can smooth the noise information our while also keeping the semantic information in the original image.

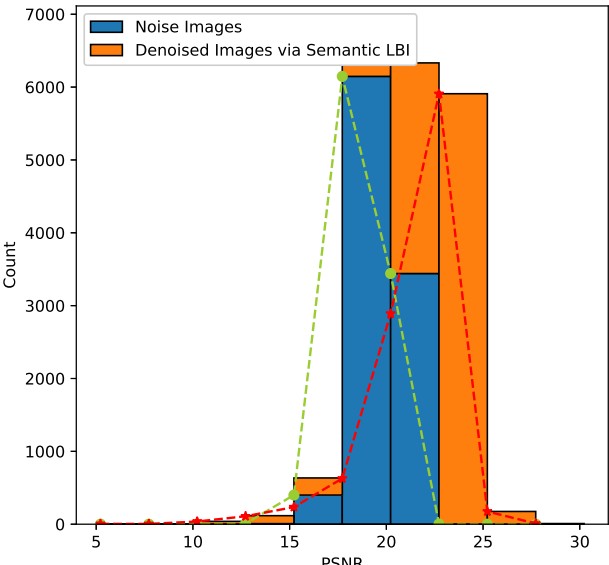

Figure 15: Histogram of PSNR of the noisy images from CIFAR10 and those after preprocessing. The blue bars denote the noise images, and the orange bars denote the denoised images via our instance smoothing algorithm.

| Samples | Preprocess | 1 | 2 | 3 | 4 | 5 | 6 | 7 | 8 | 9 | 10 | **Average** |
|---|---|---|---|---|---|---|---|---|---|---|---|---|
| std=0.01 | None | 56.0% | 52.0% | 62.0% | 54.0% | 56.0% | 57.0% | 55.0% | 52.0% | 52.0% | 61.0% | 55.7% |
| | Sparsity 0.6 | 70.0% | 73.0% | 74.0% | 67.0% | 72.0% | 71.0% | 72.0% | 65.0% | 68.0% | 78.0% | **71.0%** |
| std=0.05 | None | 48.0% | 47.0% | 49.0% | 43.0% | 49.0% | 42.0% | 47.0% | 44.0% | 43.0% | 49.0% | 46.1% |
| | Sparsity 0.6 | 63.0% | 60.0% | 62.0% | 56.0% | 59.0% | 55.0% | 69.0% | 64.0% | 62.0% | 61.0% | **61.1%** |
| std=0.1 | None | 40.0% | 38.0% | 39.0% | 33.0% | 32.0% | 38.0% | 36.0% | 35.0% | 33.0% | 33.0% | 35.7% |
| | Sparsity 0.6 | 58.0% | 56.0% | 55.0% | 48.0% | 49.0% | 53.0% | 50.0% | 54.0% | 56.0% | 45.0% | **52.4%** |

Table 12: Results of classification on noisy images from ImageNet. For each standard deviation, the 1st row corresponds to the original noisy images, while the 2nd row corresponds to our instance smoothing algorithm to preprocess test data with a sparsity of 0.6.

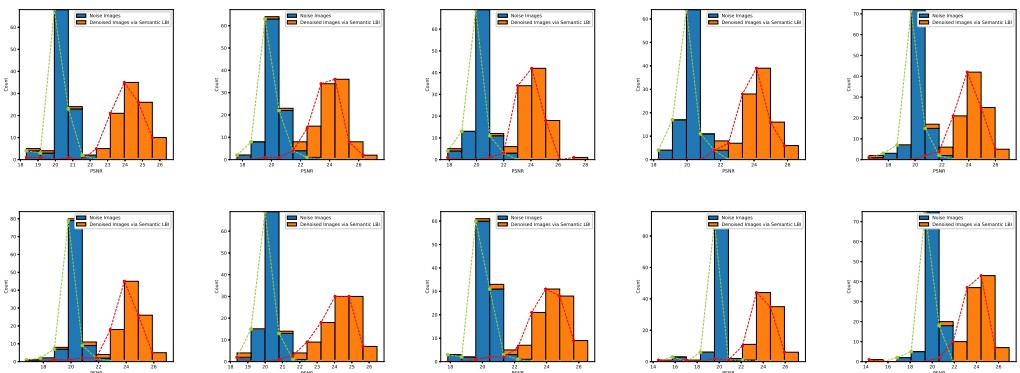

Figure 16: Histogram of PSNR of the noisy images andour instance smoothing algorithm the ones after preprocessing. Each image contains 100 samples. The blue bars denote the noise images, and the orange bars denote the denoised images via our instance smoothing algorithm.

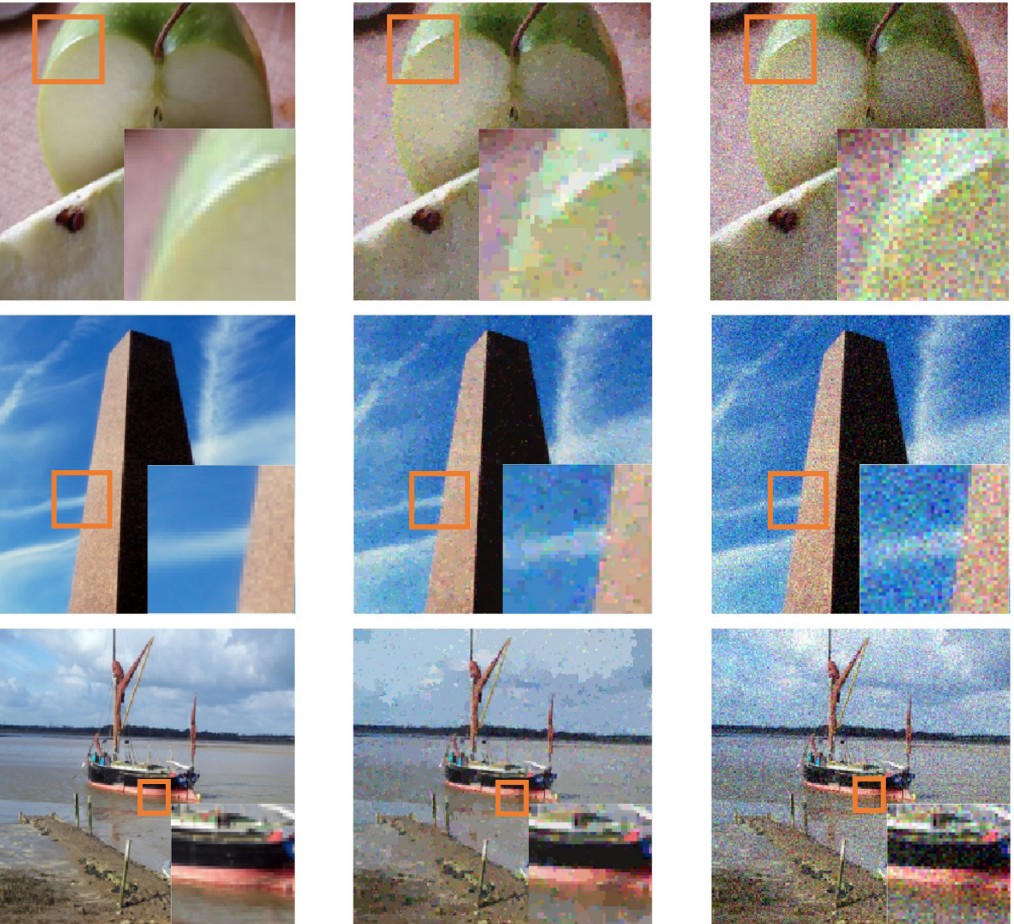

Figure 17: Examples of noisy images (right) and the denoised one via our instance smoothing algorithm (center) and original one (left). A small patch (orange box in the original image) is zoomed in for visualization.

**Comparison with TV Denoise Method.** We also compare our method with the usually used TV denoising tools. We also choose ImageNet as our dataset and use Gaussian noise with 0.1 derivation. We smooth the images with our instance smoothing algorithm in Eq. 3 to sparsity 0.6. As shown in Fig 18, our method can smooth the noise while preserving the detailed information of the images, such as color. However, TV denoise can get rid of the noise but ruin the detailed information. In Tab 13, we can also observe that the TV denoised image usually have larger variance.

| Original | Noisy | TV Denoised | Ours |
| --- | --- | --- | --- |

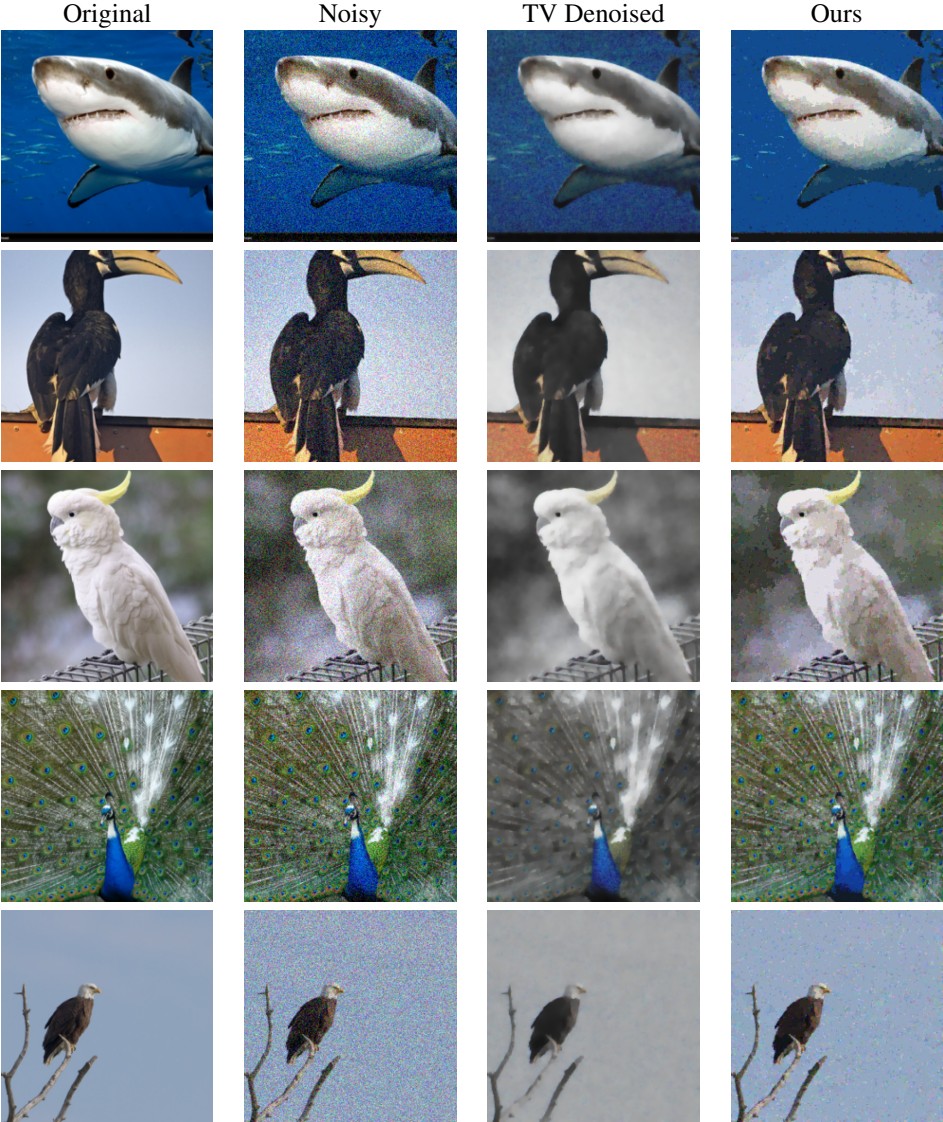

Figure 18: More denoising results comparing our Semanti-LBI method with TV denoising.

## M    LRP VISUALIZATION

The LRP Bach et al. (2015) visualization is shown in Fig. 19. It can be seen that with fix training and iterative training, the model can find the semantic part more accurately (e.g. the orange and unicycle in the first row row). When the object is found, the fix training and iterative training model tend to make decision on much bigger part of the object, especially multi-object case (e.g. the tile roof, beer bottles and dog in row 3-5). Also, the iterative training model tends to make decision on the shape of the main object related to the label (e.g. the pencil box and photocopier in row 6-7), whose decision

Table 13: The mean and variance of PSNR of 5 samples from ImageNet, comparing our method and TV denoising method.

| Method | Statics | Sample 1 | Sample 2 | Sample 3 | Sample 4 | Sample 5 |
|---|---|---|---|---|---|---|
| Noise | Mean | 20.23 | 20.30 | 20.07 | 20.20 | 20.24 |
| | Variance | 0.64 | 0.33 | 0.43 | 0.41 | 0.54 |
| TV | Mean | 23.92 | 23.95 | 23.89 | 23.83 | 23.92 |
| | Variance | 5.43 | 6.09 | 6.25 | 5.13 | 5.95 |
| Ours | Mean | 23.99 | 24.21 | 24.24 | 23.98 | 24.15 |
| | Variance | 2.36 | 2.18 | 1.31 | 1.85 | 1.31 |

pixels can capture the shape well. By finetuning on images with sparsity, the model can more focus on the object and tends to make decision on larger parts.

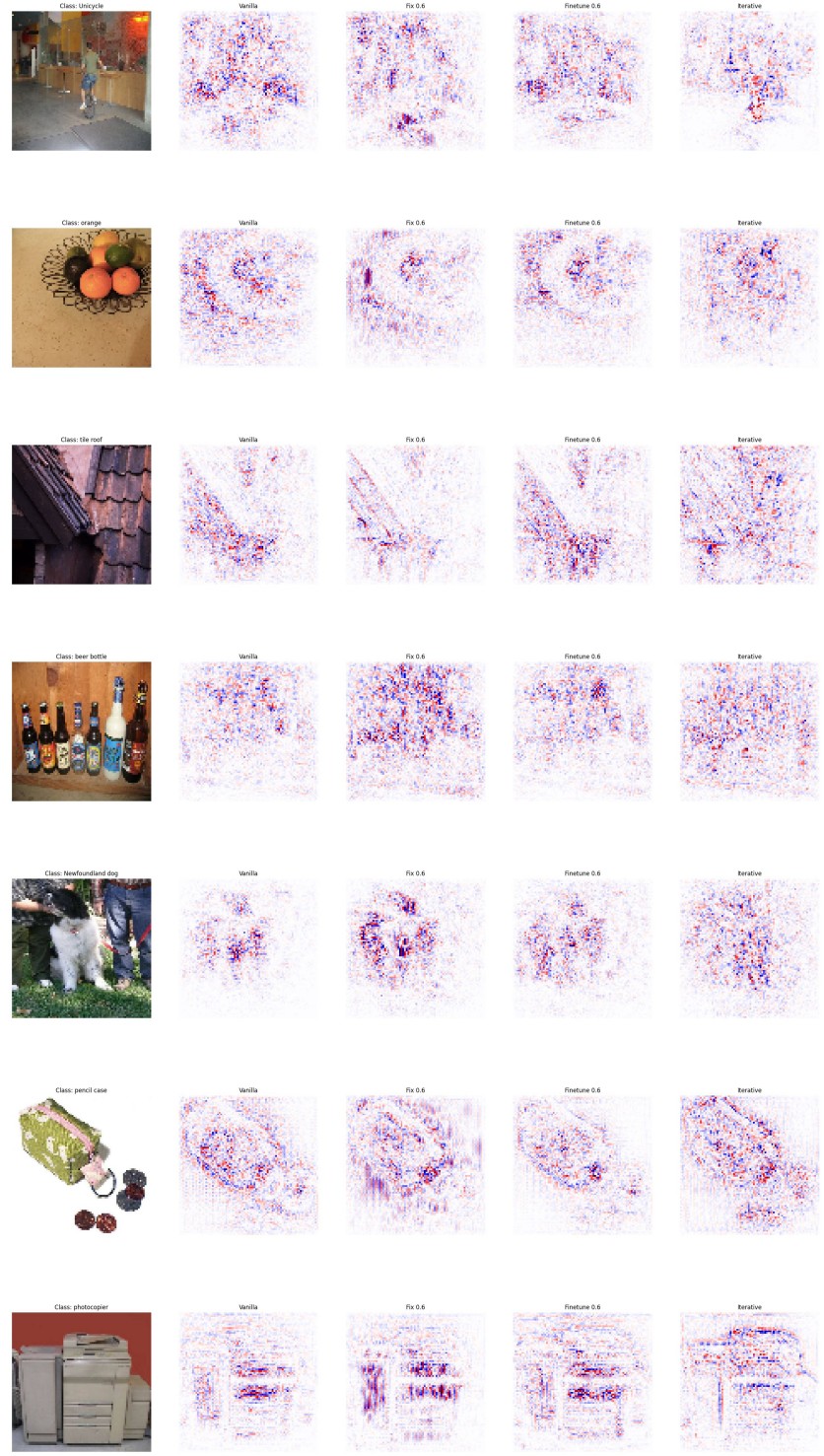

Figure 19: LRP visualization.

