# OpenReview forum: "Visual Semantic Learning via Early Stopping in Inverse Scale Space"
_ICLR.cc/2024/Conference — Submitted to ICLR 2024_

### Official Review · Reviewer_xer1 · 2023-10-30

**Soundness:** 2 fair
**Presentation:** 2 fair
**Contribution:** 2 fair
**Rating:** 5
**Confidence:** 4

**Summary:**

This work adapted “Total Variation” which has been extensively applied in areas such as image denoising for data preprocessing in neural networks, seeking to augment the network's ability to focus on low-frequency information. Through rigorous experimentation, it has been observed that this strategy markedly enhances the network's robustness, rendering it more adept at managing suboptimal input data that is either low-resolution, unclear, or highly noisy. Additionally, the approach improves the network's reliability against adversarial assaults.

**Strengths:**

1.	This paper is well-written and clear, especially with its formulas and rules, making it easy for readers to get the theory and how things are done.
2.	The paper introduces a brand new method using graph algorithms for sparse projections, which ensures the efficiency of the algorithm's execution.
3.	This approach incorporates previously extensively studied Total Variation for data preprocessing in neural networks, seeking to augment the network's ability to focus on low-frequency information.
4.	Extensive experiments prove the proposed method in dealing with substandard data, such as those of low resolution, blurriness, high noise levels and adversarial attacks.

**Weaknesses:**

1.	While this method serves as a preprocessing measure for input data, there is a notable absence of comparative validation against the performance of similar procedures (for instance, applying varying degrees of Gaussian blur or color perturbation to input images — a data augmentation approach that is, in fact, quite prevalent in network training, rather than solely introducing original clean images). The comparative analysis in Supplementary Material Fig.17 is overly subjective.
2.	Numerous studies, such as those found in reference [1,2], have delved into enhancing networks' capacity to process low-frequency information through a series of methods including image denoising and high-frequency noise injection. However, this paper lacks a comparative analysis with the findings of these studies or an exploration of whether there is room for further improvement building on their foundations.
3.	The paper proposes three training methodologies utilizing Total Variation operations, all of which yield certain results, while differing in effectiveness (for example, the method “Fixed Training” appears suitable for low-resolution scenarios, whereas “Finetune” is more applicable for adversarial attacks). However, the paper lacks an analysis explaining these variances, specifically a discussion on the applicable domains for each of the three methods.
4.	The paper lacks a discussion concerning certain parameters, especially an analysis and empirical representation of "early stopping," as mentioned in the title. Additionally, there is an absence of explanation as to why a sparsity of 0.8 was chosen for the Fixed training method, an aspect that could be elucidated with experimental demonstration and analysis
5.	The image in Fig.7 (b) is blurred. The differences in Fig.5 are not very discernible except for the first column, suggesting there's no need for so many repetitive visuals.

[1]Xie, Cihang, et al. "Feature denoising for improving adversarial robustness." Proceedings of the IEEE/CVF conference on computer vision and pattern recognition. 2019.
[2] He, Zhezhi, Adnan Siraj Rakin, and Deliang Fan. "Parametric noise injection: Trainable randomness to improve deep neural network robustness against adversarial attack." Proceedings of the IEEE/CVF Conference on Computer Vision and Pattern Recognition. 2019.

**Questions:**

It is hoped that the aforementioned issues can be addressed as comprehensively as possible. Additionally, it would be beneficial to have a more clear and intuitive description of the graph constructed in the "Sparse Projection via Graph Algorithm" section.

---

> ### Author Response · Authors · 2023-11-22
> **Response to Reviewer xer1**
>
> Thank you for your feedback and valuable suggestions regarding our work. We appreciate your comments on the clarity and novelty of our work. We address your concerns in the following.
>
>
> **Q1: While this method serves as a preprocessing measure for input data, there is a notable absence of comparative validation against the performance of similar procedures.**
>
> **A1**: Our method is not as simple as a data augmentation method, for we do not use any of the original data during our training procedure. A data augmentation method is to use noisy or corrupt image to urge the model to learn more robust information. However, our method directly discovers the most important and robust information, including structural and low-frequency information, and smooths out other suspicious information. It is much different from the data augmentation method and we do not compare the result with them.
>
>
> **Q2: Numerous studies, such as those found in reference [1,2], have delved into enhancing networks' capacity to process low-frequency information through a series of methods including image denoising and high-frequency noise injection. However, this paper lacks a comparative analysis with the findings of these studies or an exploration of whether there is room for further improvement building on their foundations.**
>
> **A2**: We have implemented [1,2] as you suggested, added comparisons with [2,3,4] in Tab.2,7, on the adversarial robustness tasks on CIFAR-10 using ResNet18. [1] performed poorly in our implementations on CIFAR-10, possibly due to its primary focus on ImageNet.
>
> It can be demonstrated that our method outperforms other methods, whether using FGSM or PGD attacks. Also, using our method as preprocessing method, our method can further improve those methods, though it is not a specially designed defence method.
>
> **Q3: Lack of analysis explaining variances among three training methodologies.**
>
> **A3**: For iterative method, it can help the model learn the structural information in the images smoothly. For Fixed method, it can learn the information in images with specific sparsity. For Finetune method, it can refine the vanilla model and help it learn more shape information.
>
> The finetune method outperform others in the adversarial robustness task because through fine-tuning, the model will learn richer information and become less susceptible to attacks, especially white-box attacks (FGSM, PGD). When a sparsity is fixed, the smooth images are very similar to low resolution images, so the fixed model trained on them performs well in low-resolution classification task. As illustrated in Fig.5, the iterative model learns low-frequency information first and then iteratively gains high-frequency information, so it can gain more stable low frequency information. Therefore, it performs well in the classification on low frequency components.
>
>
> **Q4: Lack of discussion concerning certain parameters.**
>
> **A4**: For the early stopping parameter "sparsity", in experiments we have compared the standard classification results of models trained with different sparsity levels in Tab.4, indicating that the outcome remains robust and consistent across varying levels of sparsity, provided it exceeds 0.6. This suggests a relative insensitivity to changes in
> sparsity, leading us to focus on a threshold of 0.6 for optimal balance in most of our experiments.
>
> **Q5: The image in Fig.7 (b) is blurred. The differences in Fig.5 are not very discernible except for the first column, suggesting there's no need for so many repetitive visuals.**
>
> **A5**: Thank you for pointing out the issue with Fig.7 (b). We have revised this figure for improved clarity. Regarding Fig.5, we want to show the whole training process of our iterative model, in which we can see the change in the frequency domain of feature map.
>
>
> **Q6: More clear and intuitive description of Graph algorithm.**
>
> **A6**: Thanks for the advice. We have refined this part by summarizing it in a pseudo-code format and providing an additional flowchart of the algorithm in Appendix.A. Hope these information can make it more readable.
>
> [1]Xie, Cihang, et al. "Feature denoising for improving adversarial robustness." Proceedings of the IEEE/CVF conference on computer vision and pattern recognition. 2019.
>
> [2] He, Zhezhi, Adnan Siraj Rakin, and Deliang Fan. "Parametric noise injection: Trainable randomness to improve deep neural network robustness against adversarial attack." Proceedings of the IEEE/CVF Conference on Computer Vision and Pattern Recognition. 2019.
>
> [3] Wang, Haohan, et al. "High-frequency component helps explain the generalization of convolutional neural networks." Proceedings of the IEEE/CVF conference on computer vision and pattern recognition. 2020.
>
> [4] Raymond A. Yeh, Yuan-Ting Hu, Zhongzheng Ren, and Alexander G. Schwing. Total variation
> optimization layers for computer vision. In 2022 IEEE/CVF Conference on Computer Vision and
> Pattern Recognition (CVPR), pp. 701–711, 2022a.

---

### Official Review · Reviewer_NN23 · 2023-11-03

**Soundness:** 3 good
**Presentation:** 2 fair
**Contribution:** 2 fair
**Rating:** 6
**Confidence:** 3

**Summary:**

This paper proposes an image preprocessing approach where they disentangle the semantic structure from the details of the image to avoid textual bias in deep learning. The paper proposes to leverage the structure of the Total Variation (TV) regularization matrix in the Inverse Scale Space (ISS) to generate a regularized image path from large-scale with semantic information to fine-scale with detailed information. This approach also incorporates an early stopping mechanism for the generated image path, which is computed with high efficiency using Nesterov acceleration. They demonstrate their method on various image tasks, including robustness against noise, adversarial attacks, and low-resolution images.

**Strengths:**

It is reasonable and valuable to explore how to disentangle the large-scale semantic information from the fine-scale detailed information to conduct the high-level information from the image to the neural networks.

The idea of leveraging the graph algorithm to accelerate the sparse projection is interesting.

The experiments widely demonstrate their proposed algorithm on a variety of image tasks.

**Weaknesses:**

The writing and presentation are not clear enough. It is not easy to follow for the reader who is not familiar with the related theory [1][2]. A more detailed background introduction is recommended to add to the Appendix. Due to the presentation issues, the connection and difference between the existing theory and the method introduced in [1][2] is not clear.

Many notations are not explained well. The role of $ \beta, \gamma $ is unclear. In Section 3.1, the claim "with t playing a similar role as 1/\lambda in Eq1." is confusing. And the definition of $||D\beta||$ seems has typo.

In experiments, the compared methods are not enough. The authors only compare with the vanilla and the TV layer methods. Some related preprocessing can filter out the high frequency, or the detailed contents should also be evaluated and compared.

[1] Huang et al. "Boosting with structural sparsity: A differential inclusion approach."



[2] Fu et al. "Exploring Structural Sparsity of Deep Networks via Inverse Scale Spaces"

**Questions:**

I believe this paper needs to be refined to fix the above issues.

---

> ### Author Response · Authors · 2023-11-22
> **Response to Reviewer NN23**
>
> Thank you for your valuable time and constructive feedback. We are glad you find our method interesting and have corrected typos and clarify explanations to enhance readability. We address your concerns in the following.
>
> **Q1: The connection and difference between the existing theory and the method introduced in [1][2] is not clear.**
>
> **A1**: In [1], a method which can generate a sparse regularization path for $l_1$ regularization is proposed. In [2], the method is used to prune neural networks by generating a sparse regularization path for network parameters. In this work, we use this similar method to generate sparse regularization path on image data, to discover and make use of the structural sparsity in them. Using it, we can smooth the image and help improve the robustness of model by training them on smooth images. Also, we propose an acceleration algorithm using Graph algorithm to make the instance smoothing algorithm tractable. More detail can be found in the introduction part.
>
> **Q2: Many notations are not explained well.The role of $\beta$, $\gamma$ is unclear. In section 3.1, the claim "with t playing a similar role as $1/\lambda$ in Eq1." is confusing. And the definition of $\Vert D\beta \Vert$ seems has typo.**
>
> **A2**: We have clarified our notations in section 3.1. $\beta$ represents the TV-regularized image obtained by minimizing the loss function in Equation 1. $\beta_t$ denotes the regularized image at the $t$th iteration step. $\gamma$ serves as the variable splitting term for this regularization to control the Total Variation of images. For further clarity, each element of $\gamma$ refers to an edge of Total Variation graph, which consists of pixels as nodes and $D$ as edge adjacency matrix. Regarding the role of $t$, as mentioned in Section 3.1, the path $\gamma_t$ transitions from sparsity to density as $t$ increases. Additionally, we have corrected the typo identified in your review.
>
> **Q3: In experiments, the compared methods are not enough.**
>
> **A3**: Addition to TV-layer [3], we've added comparisons with PNI [4] and the best results presented in [5], which involve smoothing the kernel (filter out high frequency components) and adversarial training to improve the robustness.
> It can be seen in Tab.2,7 that with FGSM attack, our method can outperform them without preprocessing. With PGD attack, our method can outperform them with preprocessing with a large gap when the attack strength is large.
>
> [1] Huang et al. "Boosting with structural sparsity: A differential inclusion approach."
>
> [2] Fu et al. "Exploring Structural Sparsity of Deep Networks via Inverse Scale Spaces"
>
> [3] Raymond A. Yeh, Yuan-Ting Hu, Zhongzheng Ren, and Alexander G. Schwing. Total variation
> optimization layers for computer vision. In 2022 IEEE/CVF Conference on Computer Vision and
> Pattern Recognition (CVPR), pp. 701–711, 2022a.
>
> [4] He, Zhezhi, Adnan Siraj Rakin, and Deliang Fan. "Parametric noise injection: Trainable randomness to improve deep neural network robustness against adversarial attack." Proceedings of the IEEE/CVF Conference on Computer Vision and Pattern Recognition. 2019.
>
> [5] Wang, Haohan, et al. "High-frequency component helps explain the generalization of convolutional neural networks." Proceedings of the IEEE/CVF conference on computer vision and pattern recognition. 2020.

---

### Official Review · Reviewer_5hKV · 2023-11-07

**Soundness:** 2 fair
**Presentation:** 2 fair
**Contribution:** 2 fair
**Rating:** 5
**Confidence:** 2

**Summary:**

This paper proposes an instance smoothing algorithm that disentangles structural information from detailed ones via early stopping on generating a regularized image path from large-scale to fine-scale. Then different training procedures are proposed to incorporate this algorithm into the training process. Extensive experiments are conducted on robustness tasks to verify the effectiveness of the proposed model.

**Strengths:**

1.	This paper is the first to investigate the inverse-scale-space (ISS) property at the image level.
2.	The experiments section provides convincing visualization and frequency analysis.

**Weaknesses:**

1.	The symbols in formulas need to be specified, e.g., the meaning of β.
2.	The authors claim that they propose an efficient sparse projection method. In addition to superiority in computation and time complexity compared with SVD and LSQR, is there any advantage for performance improvement?
3.	The choice of early stopping time is unclear. Since early stopping is an important operation to disentangle structural information from detailed ones, it’s crucial to illustrate the choice of early stopping time.
4.	The comparison between this method and existing methods is missing. The authors only compare their method with baseline ones, i.e., Vanilla Model and TV Layer.

**Questions:**

Please refer to weaknesses.

---

> ### Author Response · Authors · 2023-11-22
> **Response to Reviewer 5hKV**
>
> We express our gratitude to the reviewer for your efforts and valuable insights provided for our work.  We are glad that you appreciate our novelty, and we address your other concerns below.
>
> **Q1: The symbols in formulas need to be specified.**
>
> **A1**: We have improved the explanations for symbols in section 3.1. Specifically, $\beta$ represents the dense parameter for the loss function in Eq. 1. When $\beta$ minimizes the loss, it represents the TV-regularized image. Meanwhile, $\tilde{\beta}$ in Eq. 4 denotes the sparse TV-regularized image derived by projecting $\beta$ onto the subspace formed by the support set of $\gamma$.
>
> **Q2: Other improvements of our projection method compared with SVD and LSQR.**
>
> **A2**: Our main contribution is to propose the instance smoothing method, and the efficient sparse projection method is to make the time cost tractable. For performance, since our method can derive the analytical solution just as SVD, so theoretically no difference, even results in smaller loss in practice due to numerical issue. LSQR may struggle with convergence speed or precision, relying on iterative solutions.
>
>
> **Q3: The choice of early stopping time.**
>
> **A3**: In experiments, we have tried several sparsity levels as the criterion for early stopping, and compared the standard classification results in Tab.4, indicating that the outcome remains robust and consistent across varying levels of sparsity, provided it exceeds 0.6. This suggests a relative insensitivity to changes in sparsity, leading us to focus on a threshold of 0.6 for optimal balance in most of our experiments.
>
> **Q4: The comparison between this method and existing methods is missing.**
>
> **A4**: TV-layer [1] is a TV regularization method in which we have compared our methods in Tab.1, Tab.2. We also provide new results with [2,3] in Tab.2,7 in the adversarial robustness task. It can be seen that our method can outperform it in adversarial robustness tasks with FGSM and PGD attacks on the CIFAR10 dataset. Also, as a preprocessing method, our method can further improve the accuracy of adversarial images. These results show the effectiveness of our method, though it is not a specially designed defense method. Additionally, we have incorporated LRP [4] visualizations in Appendix M.
>
>
> [1] Raymond A. Yeh, Yuan-Ting Hu, Zhongzheng Ren, and Alexander G. Schwing. Total variation
> optimization layers for computer vision. In 2022 IEEE/CVF Conference on Computer Vision and
> Pattern Recognition (CVPR), pp. 701–711, 2022a.
>
> [2] He, Zhezhi, Adnan Siraj Rakin, and Deliang Fan. "Parametric noise injection: Trainable randomness to improve deep neural network robustness against adversarial attack." Proceedings of the IEEE/CVF Conference on Computer Vision and Pattern Recognition. 2019.
>
> [3] Wang, Haohan, et al. "High-frequency component helps explain the generalization of convolutional neural networks." Proceedings of the IEEE/CVF conference on computer vision and pattern recognition. 2020.
>
> [4] Bach S, Binder A, Montavon G, Klauschen F, Müller K-R, Samek W (2015) On Pixel-Wise Explanations for Non-Linear Classifier Decisions by Layer-Wise Relevance Propagation. PLoS ONE 10(7): e0130140. https://doi.org/10.1371/journal.pone.0130140

---

### Official Review · Reviewer_bFfP · 2023-11-08

**Soundness:** 3 good
**Presentation:** 3 good
**Contribution:** 3 good
**Rating:** 6
**Confidence:** 3

**Summary:**

This article proposes a novel method for visual semantic learning based on early stopping in inverse scale space. The method can disentangle structural information from detailed information in images, and incorporate it into neural network training. The method improves the robustness and explainability of the models on various tasks, such as noisy images, adversarial attacks, and low-resolution images.

**Strengths:**

1、The paper proposes a novel instance smoothing algorithm that can disentangle semantic and detailed information in images using Total Variation regularization and Inverse Scale Space. This is a creative combination of existing ideas from image processing and sparse recovery. The paper also applies this algorithm to neural network training, which is a new domain for this kind of technique.

2、The paper provides theoretical analysis and empirical evidence to support the effectiveness and efficiency of the proposed algorithm. The paper also compares the algorithm with several baselines and demonstrates its advantages in various robustness tasks, such as noisy images, adversarial attacks, and low-resolution images.

3、The paper is well-written and organized, with clear definitions, notations, and explanations.

**Weaknesses:**

1、The authors have not validated the effectiveness and scalability of their method on larger datasets, such as Imagenet.

2、The authors have not explored the possibility of applying TV regularization on feature maps, which may further improve the robustness and explainability of the models.

3、The authors have not compared with other structure-based methods, such as shape-biased models or edge detection-based models.

4、The authors have not conducted a sensitivity analysis on different TV regularization parameters, which may affect the performance and results of the instance smoothing algorithm.

**Questions:**

Here are some concerns and questions that I have for the authors:

1、How do you choose the optimal sparsity level for different tasks and datasets? Is there a general criterion or guideline for selecting the sparsity parameter?

2、How do you compare your method with other methods that also use TV regularization or other forms of regularization to enhance robustness and interpretability, such as TVM (Yeh et al., 2022b) or LRP (Bach et al., 2015)?

3、How do you evaluate the quality and diversity of the generated image path? Do you have any quantitative or qualitative measures to show the trade-off between structural and detailed information along the path?

4、How do you handle the cases where the structural information is not sufficient or reliable for the task, such as when the shape is distorted or occluded by noise or other objects? Do you have any strategies to incorporate other sources of information, such as texture or context, to improve the performance?

---

> ### Author Response · Authors · 2023-11-22
> **Response to Reviewer bFfp**
>
> Thank you for your efforts and positive assessment of our paper. We address your concerns in the following.
>
> **Q1: Validation on larger datasets.**
>
> **A1:** We have extended our training to include ImageNet100, with the results presented in Tab.5 and achieving consistent results in standard classification.
>
> **Q2: Applying TV regularization on feature maps.**
>
> **A2:** Exploring TV regularization on feature maps is one of our future goals. To date, preliminary trials using TV regularization with our method on feature maps have been made. We train ResNet18 on CIFAR-10 with TV regularization on feature map of the first convolution layer. Evaluate the model using the same experiments in Sec 4.3, we get the results in the following table. As illustrated, the model with TV regularization on the feature map exhibits higher accuracy in capturing low-frequency components while demonstrating lower accuracy in high-frequency components. This observation suggests its capability to effectively learn low-frequency information.
>
> | Model                            | high frequency component | low frequency component |
> |------------------|---------------------------|--------------------------|
> | Vanilla                          | 30%                      | 43%                     |
> | TV regularization on feature map | 20%                      | 65%                     |
>
> **Q3: Comparison with other structure-based/TV regularization methods, such as TVM (Yeh et al., 2022b) or LRP (Bach et al., 2015)?**
>
> **A3:** We have compared our method with TVM [1] in Tab.1,2 (named as "TV layer") for noisy and adversarial robustness tasks. Our approach applies TV regularization to image data, distinguishing it from TVM, which requires alterations to the model's structure. For further comparison, we have added comparative analyses with [2,3] in Tab.2,7, focusing on adversarial robustness tasks using CIFAR-10. Additionally, we have incorporated LRP [4] visualizations in Appendix M, as per your suggestion.
>
> **Q4: Sensitivity analysis on different TV regularization parameters.**
>
> **A4:** We've compared the standard classification results of models trained with different sparsity levels in Tab.4, indicating that the outcome remains consistent across different levels of sparsity, particularly when it exceeds a threshold of 0.6.
>
> **Q5: How do you choose the optimal sparsity?**
>
> **A5:** We clarify that our study did not specifically aim to identify the optimal sparsity for each task. Our method has shown robust performance across various sparsity levels, indicating that the experimental outcomes we observed are largely independent of the chosen sparsity. While most of our experiments were conducted with sparsity settings of 0.6 and 0.8, we anticipate similar results with other levels, such as 0.7.
>
> **Q6: How do you evaluate the quality and diversity of the generated image path?**
>
> **A6:** Qualitatively, the visualizations of image paths in Fig.1 and Fig.9 illustrate the trade-off between structural and detailed information. With lower sparsity, structural information, such as shape, dominates. While with higher sparsity, more detailed information such as texture shows up.  Quantitatively, we observed that higher sparsity correlates with enhanced accuracy in standard classification tasks, while our experiments in section 4 reveal a decrease in robustness, indicating the trade-off between them. With incomplete information such the lack of detailed information when sparsity is low, the model cannot achieve high accuracy. With excessive detail learned, as in the vanilla model, the model lacks robustness.
>
> **Q7: How do you handle the cases where the structural information is not sufficient or reliable for the task?**
>
> **A7:** Sections 4.1, 4.2, and 4.4 detail experiments demonstrating the resilience of our approach in these challenging conditions with noisy images, adversarial attacks, or low-resolution data. It is shown that structural information is robust in these cases and can help improve the model's robustness.
>
>
> [1] Raymond A. Yeh, Yuan-Ting Hu, Zhongzheng Ren, and Alexander G. Schwing. Total variation
> optimization layers for computer vision. In 2022 IEEE/CVF Conference on Computer Vision and
> Pattern Recognition (CVPR), pp. 701–711, 2022a.
>
> [2] He, Zhezhi, Adnan Siraj Rakin, and Deliang Fan. "Parametric noise injection: Trainable randomness to improve deep neural network robustness against adversarial attack." Proceedings of the IEEE/CVF Conference on Computer Vision and Pattern Recognition. 2019.
>
> [3] Wang, Haohan, et al. "High-frequency component helps explain the generalization of convolutional neural networks." Proceedings of the IEEE/CVF conference on computer vision and pattern recognition. 2020.
>
> [4] Bach S, Binder A, Montavon G, Klauschen F, Müller K-R, Samek W (2015) On Pixel-Wise Explanations for Non-Linear Classifier Decisions by Layer-Wise Relevance Propagation. PLoS ONE 10(7): e0130140. https://doi.org/10.1371/journal.pone.0130140

---

### Author Response · Authors · 2023-11-22
**General Response to All Reviewers**

Dear Chairs and all Reviewers,

We would like to express our gratitude to all the reviewers for their valuable comments and efforts. We are pleased to receive feedback recognizing the novelty of our instance smoothing algorithm (Reviewer bFfp), the first of its kind to investigate the inverse-scale-space (ISS) property at the image level (Reviewer 5hKV), as well as its reasonableness and value (Reviewer NN23). We are also appreciative of the compliments on the quality of our writing (Reviewers bFfP and xer1).

The major changes are as follows:

1. We've updated Tab.2-3 in the main paper and Tab.7-8 in the appendix adding PGD attack and more comparisons with other methods, including PNI [1] suggested by xer1.

2. We've added Tab.5 in appendix.B with standard classification results on ImageNet-100 [2], as suggested by bFfP. These results are consistent with those obtained on CIFAR-10.

3. We've refined our notations including $\beta$ and $\gamma$ in section 3.1 for clearer explanations, as suggested by 5hKV and NN23.

4. We've added LRP [3] visualizations in Appendix.M, as suggested by bFfP.

5. We've detailed the pseudo-code and flowchart in appendix.A to have a more clear and intuitive description of our graph algorithm, as suggested by xer1.

6. We've refined the Fig.7 (b) to remove the blur.

 [1] He, Zhezhi, Adnan Siraj Rakin, and Deliang Fan. "Parametric noise injection: Trainable randomness to improve deep neural network robustness against adversarial attack." Proceedings of the IEEE/CVF Conference on Computer Vision and Pattern Recognition. 2019.

 [2] Deng, Jia, Wei Dong, Richard Socher, Li-Jia Li, Kai Li, and Li Fei-Fei. "ImageNet: A large-scale hierarchical image database." Proceedings of the 2009 IEEE Conference on Computer Vision and Pattern Recognition. 2009.

[3]Bach S, Binder A, Montavon G, Klauschen F, Müller K-R, Samek W (2015) On Pixel-Wise Explanations for Non-Linear Classifier Decisions by Layer-Wise Relevance Propagation. PLoS ONE 10(7): e0130140. https://doi.org/10.1371/journal.pone.0130140

---

### Meta-Review · Area_Chair_TqiR · 2023-12-08

**Metareview:**

This paper presents a novel instance smoothing algorithm, utilizing Total Variation regularization and Inverse Scale Space, enhancing neural network robustness and explainability. While the paper is well-structured and the proposed method is theoretically sound, there are concerns about the lack of scalability validation on larger datasets, comparisons with other structure-based methods, and a sensitivity analysis on regularization parameters. The authors are also urged to clarify the choice of early stopping time.

This is a borderline work. AC reads the paper, reviews, and author-reviewer discussion. AC noted the Reviewer xer1's concerns are indeed essential and unfortunately, they were not addressed. AC believes that the paper still makes a valuable contribution and if revised properly, it should be accepted soon. AC regrets to recommend reject this time.

**Justification For Why Not Higher Score:**

The paper is a borderline paper with weaknesses outweighing strengths. After rebuttal, one reviewer is not satisfied with the response.

**Justification For Why Not Lower Score:**

N/A.

---

### Decision · Program_Chairs · 2024-01-16

Reject